# Microtubule number and length determine cellular shape and function in *Plasmodium*

Benjamin Spreng[1], Hannah Fleckenstein[1,†], Patrick Kübler[1,†] (ID), Claudia Di Biagio[1,†] (ID), Madlen Benz[1,†], Pintu Patra[2,†] (ID), Ulrich S Schwarz[2] (ID), Marek Cyrklaff[1] & Friedrich Frischknecht[1,*] (ID)

## Abstract

Microtubules are cytoskeletal filaments essential for many cellular processes, including establishment and maintenance of polarity, intracellular transport, division and migration. In most metazoan cells, the number and length of microtubules are highly variable, while they can be precisely defined in some protozoan organisms. However, in either case the significance of these two key parameters for cells is not known. Here, we quantitatively studied the impact of modulating microtubule number and length in *Plasmodium*, the protozoan parasite causing malaria. Using a gene deletion and replacement strategy targeting one out of two α-tubulin genes, we show that chromosome segregation proceeds in the oocysts even in the absence of microtubules. However, fewer and shorter microtubules severely impaired the formation, motility and infectivity of *Plasmodium* sporozoites, the forms transmitted by the mosquito, which usually contain 16 microtubules. We found that α-tubulin expression levels directly determined the number of microtubules, suggesting a high nucleation barrier as supported by a mathematical model. Infectious sporozoites were only formed in parasite lines featuring at least 10 microtubules, while parasites with 9 or fewer microtubules failed to transmit.

**Keywords** cell morphogenesis; infection; malaria; microtubules; sporozoite
**Subject Categories** Cell Adhesion, Polarity & Cytoskeleton; Development & Differentiation; Microbiology, Virology & Host Pathogen Interaction
The EMBO Journal (2019) 38: e100984

## Introduction

Microtubules are cytoskeletal filaments formed as hollow cylinders from dimers of α-tubulin and β-tubulin (Olmsted & Borisy, 1973; Fojo, 2008). Microtubules can nucleate spontaneously or from a template such as the γ-tubulin ring complex (Roostalu & Surrey, 2017; Wu & Akhmanova, 2017) and can undergo phases of rapid growth and shrinkage (Desai & Mitchison, 1997; Goodson &

Jonasson, 2018). Eukaryotic cells can arrange microtubules into different assemblies, including those forming axonemes of flagella and cilia, spindles for genome segregation during cell division, or cytoplasmic asters originating from the microtubule organizing centre and mediating intracellular transport and force distribution (Fojo, 2008). Microtubules provide an attractive target for drugs that interfere with these dynamic assemblies and which are used to treat cancer and infections by pathogenic worms (Jordan & Wilson, 2004; Fennell *et al*, 2008). In some organisms, different isoforms of tubulin are expressed in different cells or tissues, suggesting that the different physical and biological properties of microtubules can be encoded in the subtle variations of the protein sequence of these isoforms (Panda *et al*, 1994; Hutchens *et al*, 1997; Ludueña & Banerjee, 2008; Sirajuddin *et al*, 2014). Cells can contain from just a few to many hundreds of microtubules (Aikawa, 1971; Osborn & Weber, 1976). While the number of microtubules in axonemes is fixed and their length is stable (Linck *et al*, 2014), these two key parameters are rarely investigated in cytoplasmic or spindle microtubules. Work in neurons showed that the number of cytoplasmic microtubule is different in neurite-forming axons or dendrites (Yu & Baas, 1994), with the axon containing about 10-fold more microtubules. However, individual microtubule length varied over three orders of magnitude. Owing to those large variations, the precise number and length of cytoplasmic microtubules are rarely taken into account or even investigated in most mammalian or model cells under study. Also, the high rate of growth and shrinkage of microtubules (Mitchison & Kirschner, 1984; Goodson & Jonasson, 2018) make the determination of these parameters difficult if not meaningless as they constantly change. Work on spindle microtubules shows that their number depends on the numbers of kinetochores (Nannas *et al*, 2014). Both, number and length of spindle microtubules, can be fixed and are likely important for spindle function, as shown in fission yeast (Ward *et al*, 2014).

Similar to metazoans, protozoans can also contain unique assemblies of microtubules, such as those making up the suction disc of the intestinal parasite *Giardia lamblia* (Nosala *et al*, 2018) or the axopodia of the free-living heliozoan *Echinosphaerium nucleofilum* (Tilney & Porter, 1965). Key protozoan organisms that are intensely investigated for their diverse biology and medical relevance include

1 Integrative Parasitology, Center for Infectious Diseases, Heidelberg University Medical School, Heidelberg, Germany
2 Institute for Theoretical Physics and Bioquant, Heidelberg University, Heidelberg, Germany
*Corresponding author. Tel: +49 6221 566537; Fax: +49 6221 564643; E-mail: freddy.frischknecht@med.uni-heidelberg.de
†These authors contributed equally to this work

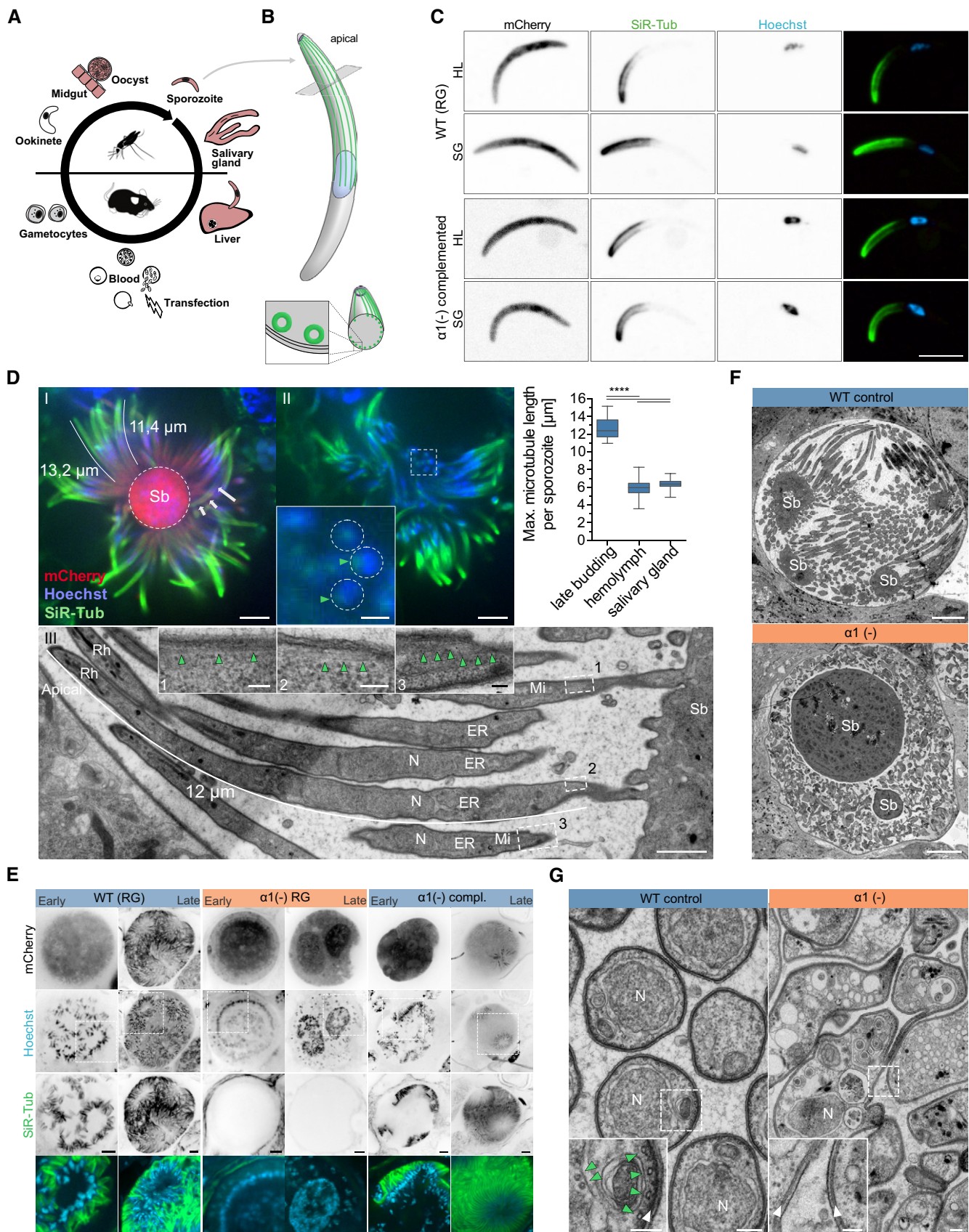

Figure 1.

◀ **Figure 1.** α1-tubulin is essential for sporozoite formation.

A   Cartoon of a simplified *Plasmodium* life cycle indicating in red the stages important for our work. Note that transfection is performed in blood-stage parasites.

B   Schematic cartoon of a sporozoite with an indicated apical cross section. Subpellicular (cytoplasmic) microtubules (green) can be found below the three membrane layers (plasma membrane and inner membrane complex) of the pellicle (bottom).

C   Isolated sporozoites from the hemolymph (HL) and salivary glands (SG) of infected mosquitoes stained with Hoechst and SiR-tubulin. Note the similar staining of WT and α1-tubulin(-) complemented lines. Scale bar: 5 μm.

D   SiR-tubulin staining (green, I and II) and TEM (III) show that microtubules shrink after wild-type sporozoite formation. During budding, microtubules can be up to 15 μm long (I), while after formation they measure on average only 6 μm (graph). Inset in II shows sporozoite cross section at the sporoblast membrane, note the SiR-tubulin fluorescence (green, arrowheads) next to the nuclear Hoechst stain (blue). (III) TEM longitudinal section of budding sporozoites with the length of a subpellicular microtubule indicated by a white line. Green arrowheads point to microtubules close to the sporoblast. Rh: rhoptry; N: nucleus; ER: endoplasmic reticulum; Mi: mitochondrion; Sb: sporoblast. **** indicates *P* < 0.0001; Kruskal–Wallis test. Scale bars: I and II: 5 μm, magnification box in II: 1 μm, III: 1 μm, magnification boxes in III: 0.1 μm.

E   Comparison of sporozoite development in oocysts of WT, α1-tubulin(-) and complemented lines. Hoechst (blue) and SiR-tubulin (green) staining reveal that the complemented line generates sporozoites indistinguishable from WT controls, while SiR-tubulin staining is absent from α1-tubulin(-) oocysts. mCherry stains the cytoplasm of the parasite. Scale bars: 5 μm.

F   Transmission electron micrographs of developing sporozoites in WT and α1-tubulin(-) parasite lines. Note the slender sporozoites in the WT oocysts, which are absent in the mutant. Sb: sporoblast; scale bars: 5 μm. See also Appendix Figs S2 and S3.

G   TEM cross sections of WT and α1-tubulin(-) parasites showing the aberrant structure of the latter. Subpellicular and centriolar plaque (spindle pole body)-associated (hemi-spindle) microtubules are readily recognizable in the WT sporozoites (green arrowheads in insert) but absent in the mutant, while the inner membrane complex is formed below the plasma membrane of both parasites (white arrowheads). Note the different scale chosen to highlight the malformed shapes in the mutant. N: nucleus; scale bars: 0.2 μm, magnification boxes: 0.1 μm.

Source data are available online for this figure.

the parasites *Trypanosoma brucei* causing sleeping sickness and various species of *Plasmodium*, the causative agents of malaria. Both contain arrays of so-called subpellicular (cytoplasmic) microtubules located in close contact to the surface (pellicle) of the parasites (Garnham *et al*, 1960, 1961, 1962, 1963; Vanderberg *et al*, 1967; Sinden & Garnham, 1973; Gull, 1999; Kappes & Rohrbach, 2007; Lacomble *et al*, 2009), although these parasites are found on different branches on the eukaryotic tree of life.

Here, we studied the impact of the subpellicular microtubules on the formation and function of the form of *Plasmodium* transmitted by mosquitoes using the rodent model species *Plasmodium berghei*. These so-called *Plasmodium* sporozoites develop in oocysts at the midgut wall of mosquitoes and enter the salivary glands of the insect from where they are transmitted to a vertebrate host (Ménard *et al*, 2013; Frischknecht & Matuschewski, 2017; Fig 1A). Sporozoites are highly flexible and motile cells that need to pass several tissue barriers including the salivary glands, the skin and blood vessel endothelium before they infect liver cells to differentiate into thousands of merozoites, which then infect red blood cells (Frischknecht & Matuschewski, 2017; Vaughan & Kappe, 2017). Sporozoites are the target of the first approved malaria vaccine, which contains part of the major sporozoite surface protein as an antigen (Clemens & Moorthy, 2016; Olotu *et al*, 2016). They are also used in ongoing clinical trials as live vaccines in combination with blood-stage killing drugs or as attenuated parasites (Matuschewski, 2017; Singer & Frischknecht, 2017). Sporozoites of *P. berghei* contain a fixed number of 16 subpellicular microtubules that extend from an apical polar ring at the front of the sporozoite to the nucleus at the centre of the cell (Vanderberg *et al*, 1967; Kudryashev *et al*, 2010, 2012; Fig 1B). These microtubules are arranged in a typical 1 + 15 pattern (Fig 1B) and have been suggested to be important for vesicular trafficking, morphogenesis, cellular mechanics, polarity and motility of sporozoites (Vanderberg *et al*, 1967; Cyrklaff *et al*, 2007; Schrevel *et al*, 2007). The subpellicular microtubules are considered extremely stable and cannot be depolymerized by classic microtubule depolymerizing agents, while addition of taxol derivatives during replication of parasites showed the importance of microtubules in the *Plasmodium* blood stage (Pouvelle *et al*, 1994; Schrével *et al*, 1994; Sinou *et al*, 1996; Kappes & Rohrbach, 2007).

Transgenic parasites can be generated in the blood stages of *Plasmodium* and then transmitted to mosquitoes if the genetic modification is not essential for parasite growth and development in the rodent. In order to assess microtubule function in sporozoites, we now used a gene deletion and replacement approach for *α1-tubulin* to assess the function of microtubules for sporozoite formation and the impact of modulating their numbers and length on sporozoite biology. Strikingly, these perturbations revealed that *Plasmodium* genomes can be separated without microtubules and that microtubule length and number are essential for efficient transmission from the mosquito to the mammalian host. Our work demonstrates the essential role of microtubule regulation in a small protozoan of high medical relevance and might pave the way to similar insights in other model organisms.

**Table 1.   Early sporozoite formation of the generated parasite lines.**

| Parasite line | Midgut sporozoites | | Hemolymph sporozoites | |
| --- | --- | --- | --- | --- |
| | Sporozoites / mosquito (n) | N° of apical subpellicular microtubules (n) | Subpellicular microtubule length in μm (n) | Sporozoite morphology |
| WT control | 88,000 (64) | 16 (23) | 6,0 (120) | +++ |
| α1^Δ introns | 59,200 (34) | 16 (30) | 6,1 (115) | +++ |
| α1^Δ C-term | 88,300 (57) | 16 (40) | 5,9 (69) | +++ |
| α2^+ | 49,700 (65) | 16 (33) | 4,6 (156) | ++ |
| α2^++ | 92,400 (46) | 11 (39) | 5,5 (130) | ++ |
| α2^+++ | 96,400 (34) | 9 (27) | 4,8 (130) | + |
| α1^cm, Δintrons | 59,600 (89) | 6 (27) | 6,1 (96) | - |
| α1(-) | 0 (70) | 0 (35) | n.a. | n.a. |

Green numbers indicate similar levels to wild type, orange indicates lower levels, and red indicates large deviations from the wild type.

**Table 2.  Infective capacity of sporozoites from the generated parasite lines.**

| Parasite line | Sporozoites / mosquito (n) | Salivary gland invasion ratio | Subpellicular microtubule length in μm (n) | Sporozoite length (n) | Sporozoite gliding diameter (n) | Moving sporozoites (n) | Clockwise moving sporozoites (n) | Infected mice by 10 mosquito bites / total mice (prepatency) | Infected mice by 10.000 sporozoites injected i.v. / total mice (prepatency) |
|---|---|---|---|---|---|---|---|---|---|
| | | | | | Salivary gland sporozoites | | | | |
| WT control | 20,600 (61) | 0.23 | 6,4 (122) | 12,4 (75) | 13,4 (209) | 78 % (372) | 2,4 % (291) | 7/8 (3,9) | 11/12 (3,7) |
| α1$^{\Delta\,introns}$ | 12,700 (34) | 0.21 | 6,1 (127) | 12,3 (60) | 13,4 (58) | 73 % (580) | 2,1 % (424) | 8/8 (4,4) | 7/8 (4,4) |
| α1$^{\Delta\,C\text{-}term}$ | 30,000 (53) | 0.34 | 6,2 (79) | 12,4 (81) | 11,2 (166) | 79 % (283) | 3,1 % (223) | 4/4 (4,3) | 4/4 (3,5) |
| α2$^{+}$ | 5,400 (66) | 0.11 | 4,5 (98) | 8,5 (80) | 7,2 (211) | 69 % (350) | 5,8 % (241) | 9/12 (5,8) | 7/8 (5,3) |
| α2$^{++}$ | 12,200 (47) | 0.13 | 5,2 (116) | 10,8 (97) | 11,4 (176) | 74 % (316) | 10,3 % (233) | 8/13 (4,6) | 12/12 (4,3) |
| α2$^{+++}$ | 300 (33) | 0.00 | 5,0 (69) | 9,6 (52) | 11,3 (20) | 32 % (116) | 21,6 % (37) | n.d. | n.d. |
| α1$^{cm,\,\Delta introns}$ | 6 (87) | 0.00 | n.a. | n.a. | n.a. | n.a. | n.a. | 0/4 (∞) | n.a. |
| α1(-) | 0 (67) | n.a. | n.a. | n.a. | n.a. | n.a. | n.a. | n.a. | n.a. |

Green numbers indicate similar levels to wild type, orange indicates lower levels, and red indicates large deviations from the wild type.

# Results

## Deletion of *α1-tubulin* affects *Plasmodium* sporozoite formation during budding

All sequenced *Plasmodium* genomes contain one gene encoding β-tubulin and two genes encoding α-tubulins (Fig EV1A and B). The two *α-tubulins* likely arose by gene duplication from an ancestral α-tubulin. Such duplications can allow for a change in either the level or the timing of gene expression and/or to generate different functionalities arising from differences within the two coding sequences. Both mechanisms were shown to be at work for α-tubulins in different organisms (Ludueña & Banerjee, 2008). Expression analysis suggested that the *α2-tubulin* gene is expressed between one and two orders of magnitude higher than the *α1-tubulin* gene in blood stages (Otto *et al*, 2014). Indeed, the *α2-tubulin* gene appears essential and cannot be deleted (Kooij *et al*, 2005), while we could readily delete the *α1-tubulin* gene in a wild-type (WT) background and in a parasite expressing GFP and mCherry as cytoplasmic markers at different life cycle stages (Fig EV1C and D). *α1-tubulin* gene deletion was attempted before without success (Kooij *et al*, 2005); however, the improved transfection methods since then might have allowed us to generate a transgenic line without problems. Both *α1-tubulin*(-) parasite lines showed no defect in blood-stage growth or initial mosquito infection (Fig EV1E and F). However, no sporozoites were found in the midgut or salivary glands of the mosquito (Tables 1 and 2); thus, a main defect seems to occur at the oocyst stage. This defect could be completely rescued by complementing the gene in the *α1-tubulin*(-) parasite line (Fig EV2).

To investigate whether microtubules were affected by the deletion of *α1-tubulin,* we established the use of SiR-tubulin, a derivative of taxol that only fluoresces when associated with microtubules (Lukinavičius *et al*, 2014) for sporozoite (Fig 1C) and oocyst labelling (Fig 1D and E). This showed that microtubules could readily be detected in oocysts and isolated sporozoites with spinning disc confocal microscopy and that their length could be determined with great precision (Fig 1D and Appendix Fig S1). This showed that

during late budding in the oocyst microtubules were longest and that they subsequently shrank as the sporozoites matured in the oocysts to again slightly grow as the parasites reside in the salivary gland (Fig 1D). This suggests that, contrary to previous suggestions (Cyrklaff *et al*, 2007), cytoplasmic microtubules in *Plasmodium* undergo some level of shrinkage and growth, although much more slowly than in mammalian cells, on the time scale of hours to days rather than seconds.

Labelling of nuclei with the DNA-dye Hoechst confirmed the close association of microtubules and nuclei (Fig 1C and D) as expected during the formation of wild-type sporozoites (Vanderberg *et al*, 1967; Schrevel *et al*, 2007; Appendix Fig S2). However, in the oocysts of *α1-tubulin*(-) parasites we could not detect any microtubules (Fig 1E), while they were readily visible in the complemented line (Fig 1C–E). Electron microscopy analysis further showed that no sporozoites were found in oocysts from *α1-tubulin*(-) parasites (Fig 1F and Appendix Figs S2 and S3). Detailed views of the micrographs revealed that, as in wild-type parasites, the plasma membrane was still subtended by the inner membrane complex (IMC), an organelle defining the alveolates (Harding & Meissner, 2014). However, no normally shaped parasites could be detected in *α1-tubulin*(-) oocysts. Instead, the parasites presented as undulating shapes with many vesicular structures of unclear provenance. In some sections, we could detect a nucleus enclosed in those deformed "parasites" (Fig 1G). Close inspection of the cytoplasm next to the IMC revealed that in contrast to wild-type, *α1-tubulin*(-) parasites lacked microtubules (Fig 1G).

Sporozoites form by budding from the plasma membrane surrounding the sporoblast, which contains the cytoplasm, nuclei and essential organelles (Sinden & Garnham, 1973; Schrével *et al*, 1977; Sinden & Strong, 1978; Thathy *et al*, 2002; Schrevel *et al*, 2007; Appendix Fig S2). The undulating shapes of the *α1-tubulin*(-) parasites suggested that initial budding might still proceed. To investigate this in more detail, we studied different steps of sporozoite formation in wild-type and *α1-tubulin*(-) parasites by transmission electron microscopy (TEM) and by tomographic reconstructions from serial TEM sections (Fig 2A and B). TEM revealed that sporozoite budding appears to initiate normally in *α1-tubulin*(-) parasites (Fig 2A and Appendix Figs S2 and S3). The

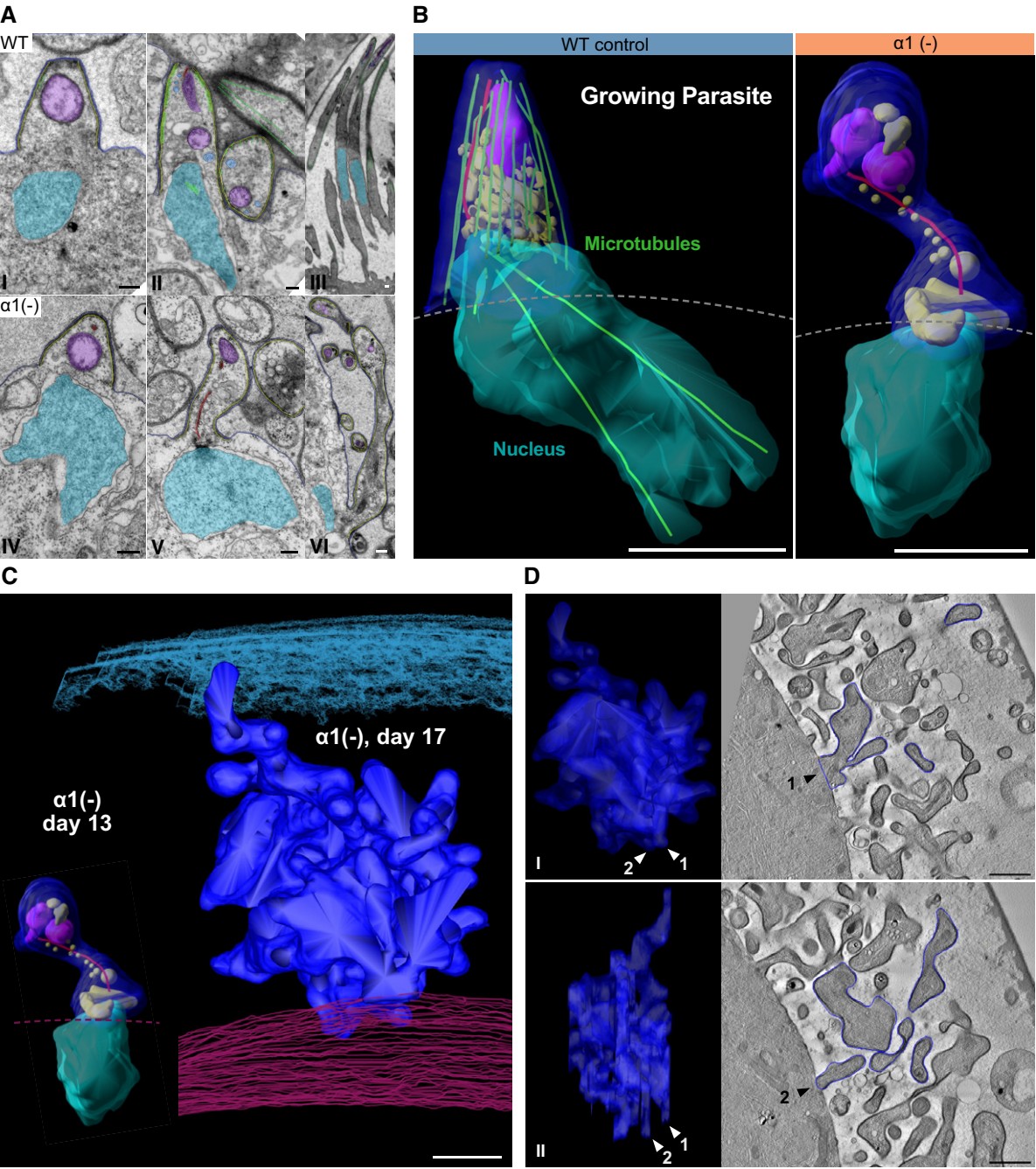

**Figure 2. Sporozoite budding is impaired in α1-tubulin(-) parasites.**

A   Electron micrographs showing different stages of development in WT and *α1-tubulin*(-) parasites. Key structural elements are highlighted in the following colours: blue: plasma membrane; yellow: inner membrane complex; green: microtubules; light blue: micronemes; lilac: rhoptries; red: rootlet fibre; and cyan: nucleus. Note the undulating shapes of the elongated mutants and that micronemes are hard to distinguish from other small vesicles in the *α1-tubulin*(-) parasites. The rootlet fibre in panel II starts at the apical tip and in panel V is associated with the centriolar plaque (dark structure in nuclear envelope). Panel III corresponds to Fig 1D. Scale bars: 200 nm.

B   3D reconstruction from serial sectioning reveals that growing *α1-tubulin*(-) "sporozoites" lack apical polarity and fail to pull in nuclei. Microtubuli: green; rootlet fibre (connecting apical pole to centriolar plaque): red; rhoptries: purple; endomembraneous vesicles (Golgi, premicronemes): brown; nucleus: cyan; sporozoite plasma membrane and inner membrane complex: blue; dotted lines show location of sporoblast membrane. 17 and 10 sections of 100 and 300 nm thickness were processed for the tomograms in WT and *α1-tubulin*(-) parasites, respectively. Scale bar: 1 μm. See also Movie EV1.

C   Tomography of serial sections from budding *α1-tubulin*(-) parasites at day 13 (same as in panel B) and day 17 highlights the increase in surface area (blue) with time. The oocyst wall at day 17 is highlighted by cyan lines (top), and the sporoblast membrane is highlighted by red lines (bottom). The sporoblast membrane at day 13 is indicated by a dashed line (red). 10 and 11 sections of 300 nm thickness were processed for the tomograms at days 13 and 17, respectively. A single continuous 'parasite' was highlighted at day 17. Scale bar: 1 μm.

D   *α1-tubulin*(-) parasites show multiple attachments (arrowheads) with the sporoblast membrane at day 17 post-infection. Note the highly branched nature of the budding parasite (same parasite as in panel C) shown in two views in I and II. Right: slices through the tomogram in grey scale with the outlines used for the 3D model indicated. Two connections with the sporoblast are indicated by 1 and 2. Scale bars: 1 μm. See also Movie EV2.

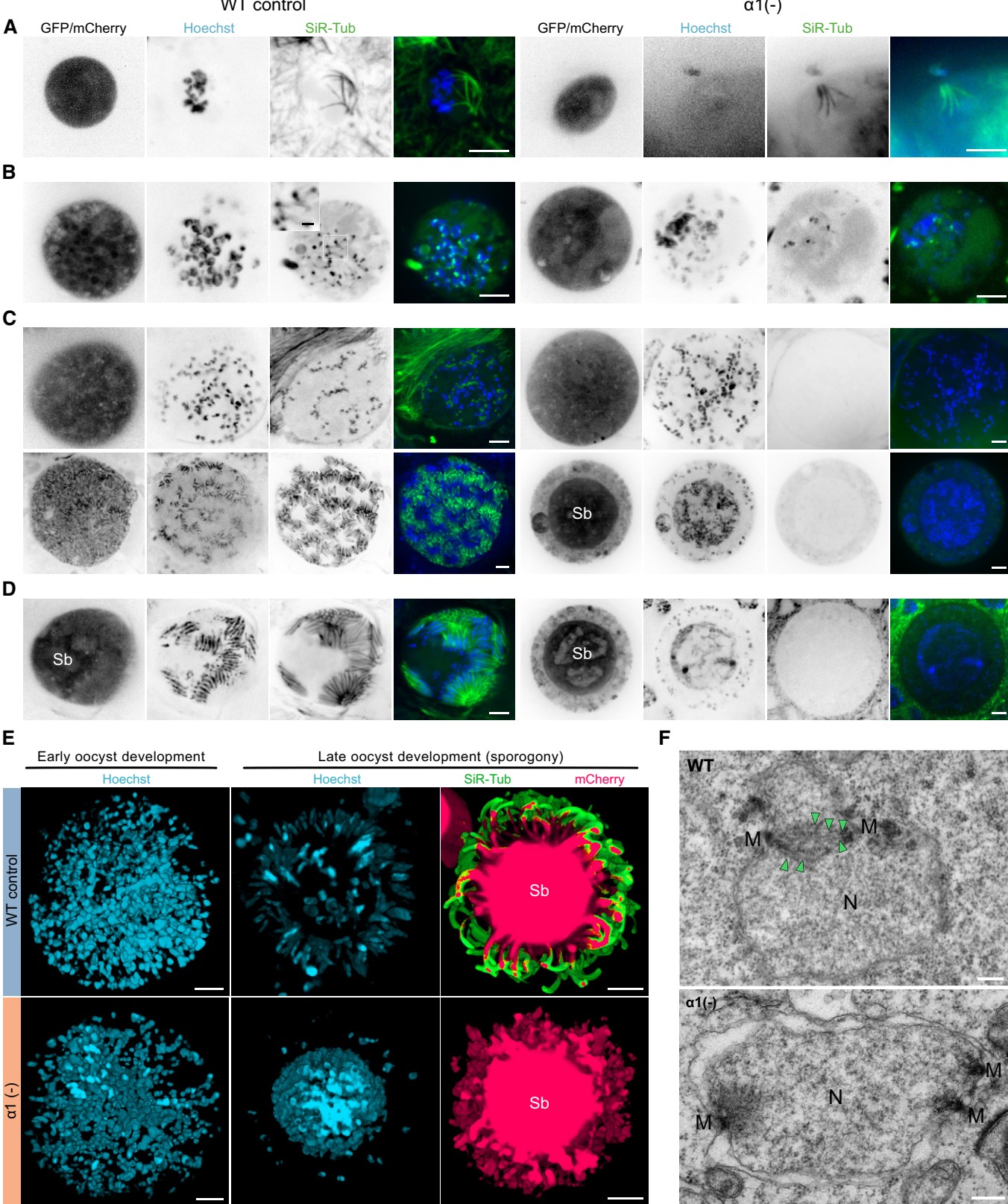

Figure 3.

**Figure 3.  Genome replication and nuclear multiplication are proceeding in the absence of microtubules.**

A–D   Live imaging of wild-type (WT) and *α1-tubulin*(-) oocysts expressing *ef1α:GFP* (early and late oocysts) and *csp:mCherry* (late oocysts) to locate the oocyst cytoplasm. Microtubules were labelled with SiR-tubulin (green) and DNA with Hoechst (blue). Oocysts are shown in a chronological order from early (day 4) to late (day 12) development. (A, B) Early oocyst development with remaining subpellicular microtubules of the preceding ookinete stage (A) and subsequent DNA replication (B), where some but not all DNA is co-stained by microtubules in the mutant. (C) Nuclear alignments to the invaginated plasma membrane and budding of sporozoites. Note the strong SiR-tubulin signal only seen in sporozoites of WT oocysts and complete absence from the mutant. Sb: sporoblast. (D) Oocyst ready to burst with fully formed sporozoites in WT oocysts. Note that *α1-tubulin*(-) oocysts retained sporozoite nuclei predominantly within the sporoblast (Sb) during budding. Scale bars: 5 μm, magnification box in (B): 1 μm.

E   3D reconstructions from spinning disc confocal microscopy z-stacks across an early oocyst labelled with the DNA-dye Hoechst (left). 50 and 19 images with a z-distance of 0.5 and 1 μm were collated for WT and *α1-tubulin*(-) oocysts, respectively. Right: 3D reconstructions from late WT and *α1-tubulin*(-) oocysts labelled with Hoechst (blue) and SiR-tubulin (green). Note that the mutant does not form proper sporozoite shapes during budding and the few peripheral nuclei that were uptaken into budding sporozoites. 10 and 28 images with a z-distance of 1.5 and 1 μm were collated for WT and *α1-tubulin*(-) oocysts, respectively. Sb: sporoblast; scale bars: 5 μm.

F   TEM cross sections of WT and *α1-tubulin*(-) oocysts during nuclear division. Spindle microtubules (arrowheads) emerging from centriolar plaques (M) were seen in WT oocysts. Although individual nuclei (N) were visible in *α1*(-) oocysts, microtubules were absent even in dividing centriolar plaques (M). Scale bars: 0.2 μm.

nuclei appeared aligned underneath the plasma membrane as in wild type with their spindle pole body (centriolar plaque) oriented towards the growing apical end of the sporozoite and possibly still connected to the growing tip by the rootlet fibre, a dynamic tether that links the centriolar plaque to the apical end of the emerging parasites (Sinden & Strong, 1978; Schrevel *et al*, 2007; Francia *et al*, 2012). However, the growing parasites in the mutants showed irregular shapes that contrasted the more regular and elongated WT sporozoite form (Fig 2A and B). While the inner membrane complex subtended the plasma membrane of both budding WT and mutant sporozoites (Fig 2A), the growing *α1-tubulin*(-) parasites were malformed and contained numerous vesicular structures (Fig 2A and B). The nuclei lacked intra-nuclear microtubules and were not elongated like those within the growing WT sporozoites, where intra-nuclear microtubules extended along the entire longitudinal axis of the ovoid nuclei (Fig 2A and B; Movie EV1). Later during development, *α1-tubulin*(-) parasites showed large extensions from the sporoblast membrane with multiple bends and diverse diameters and rarely contained nuclei (Fig 2C and D, Movie EV2). These data suggest that microtubules play an essential structural role during sporozoite formation in the oocyst and that specifically α1-tubulin is required for microtubule formation at this stage.

## *α1-tubulin* deletion does not affect genome replication and nuclear division

As the deletion of genes could lead to upregulation of other genes necessary in the affected process, we next investigated mRNA levels of *α1-tubulin* and *α2-tubulin* by qRT–PCR. This showed that little *α2-tubulin* mRNA was present during sporozoite formation (Fig EV3A) and that levels of *α2-tubulin* mRNA were unaffected by *α1-tubulin* deletion (Fig EV3B). Similarly, we also noted that the expression of the major surface antigen, circumsporozoite protein (Cohen *et al*, 2010; Frischknecht & Matuschewski, 2017), was also not affected by the deletion of *α1-tubulin* (Fig EV3C); albeit, a tendency for lower expression levels could be detected at days 12 and 14 post-mosquito infection.

We next analysed the *α1-tubulin*(-) parasite line that expressed mCherry from the *csp* promoter. This line also expresses GFP from a constitutively active promoter and thus allows detecting developing oocysts by GFP, the onset of *csp* promoter activity during parasite formation by mCherry, DNA replication by addition of Hoechst and microtubule formation by SiR-tubulin, which fluoresces in the far red (Lukinavičius *et al*, 2014). We noted that in cells where

few nuclei are present and no alignment of the nuclei to the sporoblast membrane had taken place, no mCherry and no SiR-tubulin signal could be detected (Fig EV3D). In contrast, in parasites that progressed further in their development, mCherry was strongly expressed, which coincided with the formation of microtubules (Fig EV3D). In comparison with wild-type parasites, the *α1-tubulin*(-) parasites progressed slower towards final nuclear alignment and showed fewer oocysts at the same day post-infection with high levels of mCherry expression (Fig EV3E), suggesting some developmental delay in the program governing sporozoite formation.

Strikingly, co-labelling with the DNA stain Hoechst and our TEM images consistently showed that nuclear division and alignment at the periphery of the sporoblast (Burda *et al*, 2017) appeared initially not affected in *α1-tubulin*(-) parasites (Figs 1E and 2) suggesting that chromosome segregation could proceed in the absence of microtubules, a phenomenon only reported once before for eukaryotic cells in fission yeast (Castagnetti *et al*, 2010). To investigate whether microtubules were absent in early stages of nuclear division within oocysts, we imaged oocysts from day 4 after infection. This revealed that the microtubules from ookinetes, the stages invading the mosquito midgut and transforming into oocysts could still be detected until day 5 post-infection in both wild-type and *α1-tubulin*(-) parasites (Fig 3A). With advancing genome replication and beginning nuclear division, microtubules could be detected as spots in association with only some nuclei in *α1-tubulin*(-) parasites, while nuclei of wild-type parasites showed microtubule labelling in the nuclei, clearly appearing in a hemi-spindle form (Fig 3B). Similar staining was found in replicating blood stages of both wild-type and *α1-tubulin*(-) parasites (Fig EV4A). In later oocyst stages, no microtubule signal could be found anymore in *α1-tubulin*(-) parasites confirming the above-reported absence from both cytoplasm and nucleus (Fig 3C). In contrast, many nuclei could readily be detected in *α1-tubulin*(-) parasites, although they appeared to clump within the oocyst centre and few distributed to the periphery as sporozoites started to bud. Also at times after infection when final stages of sporozoite formation were detected and wild-type sporozoites filled the oocysts, no microtubule staining was found in *α1-tubulin*(-) parasites and only few nuclei were detected in the periphery (Fig 3D and E). Lastly, investigation of dividing centriolar plaques by TEM showed the absence of nuclear microtubules in *α1-tubulin*(-) parasites from day 7 onwards, while they were readily detectable in the wild-type and in young *α1-tubulin*(-) oocysts (Figs 3F and EV4B).

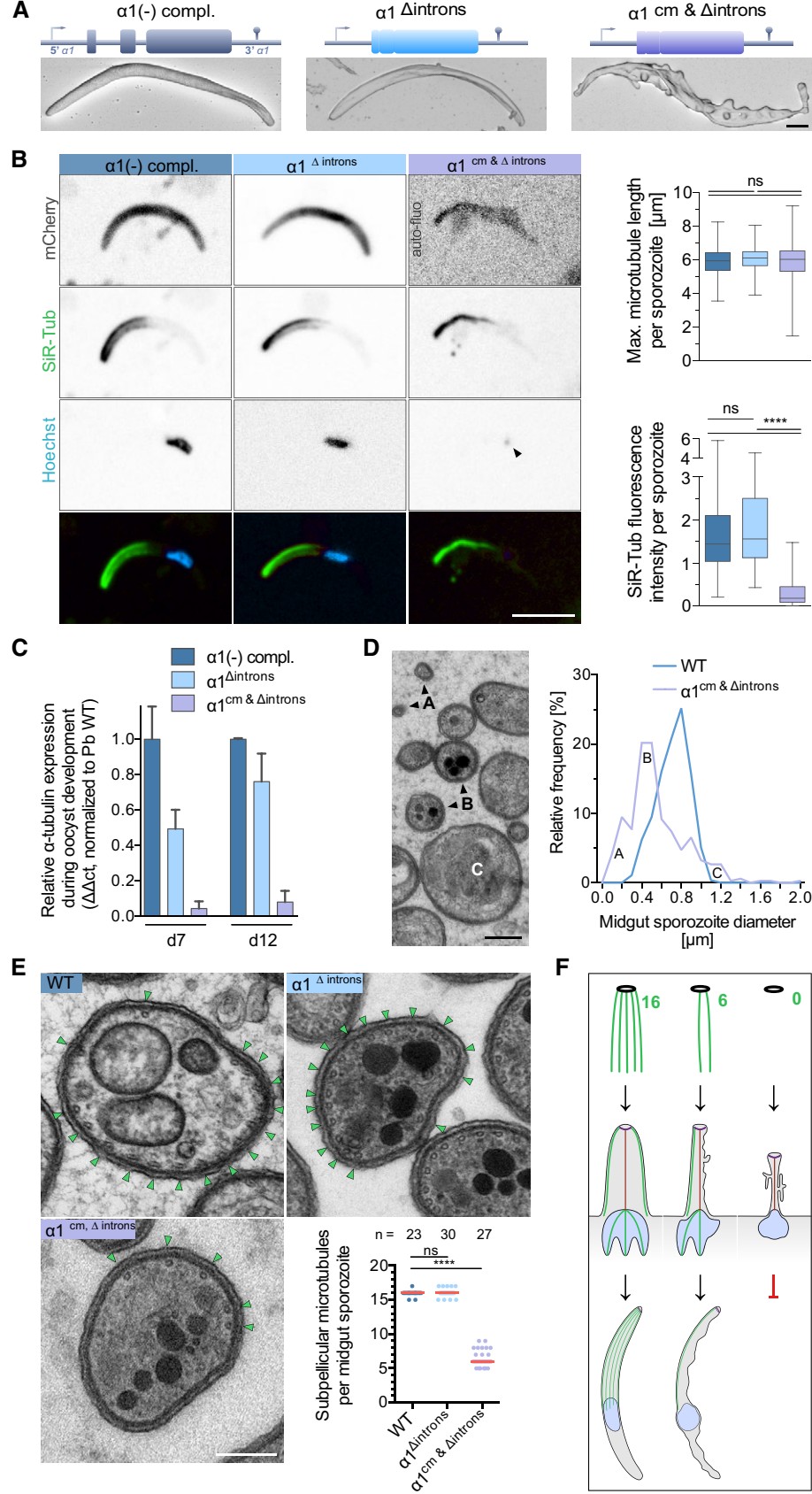

Figure 4.

◀

**Figure 4.  Lowering expression of α1-tubulin leads to fewer microtubules and aberrant sporozoite shapes.**

A  Complementation of *α1-tubulin*(-) parasites with different *α1-tubulin* constructs lacking introns (α1$^{\Delta introns}$) or featuring only codon-modified exons (α1$^{cm\&\Delta introns}$). Scanning electron micrographs show the aberrant shape of the α1$^{cm\&\Delta introns}$-expressing sporozoites. See also Fig EV5. Scale bar: 1 μm.

B  Fluorescence images showing the SiR-tubulin-stained microtubules, nuclei (Hoechst) and cytoplasmic shape (mCherry) of the parasite lines. Note the small amount of DNA (black arrowhead) and the reduced level of SiR-tubulin fluorescence in the α1$^{cm\&\Delta introns}$-expressing sporozoites (lower graph). See also Appendix Fig S6. Quantitative data were derived from 96 to 120 sporozoites per line. Box plots represent 50% (boxes) and 95% (bars) of data with horizontal bar showing the median. Scale bar: 5 μm. **** indicates *P* < 0.0001, Kruskal–Wallis test. Note that α1$^{cm\&\Delta introns}$-expressing sporozoites do not express mCherry; their autofluorescence (auto-fluo) indicates an aberrant shape.

C  qRT–PCR analysis of the three parasite lines at days 7 and 12 after infection. Error bars correspond to the standard deviation from two biological replicates.

D  TEM reveals α1$^{cm\&\Delta introns}$-expressing sporozoites with smaller (A and B) and larger (C) diameters than wild-type sporozoites. See also Appendix Fig S7. Scale bar: 0.2 μm.

E  α1$^{cm\&\Delta introns}$-expressing sporozoites show a reduced number of subpellicular microtubules (green arrowheads) by TEM. **** indicates *P* < 0.0001, ns: not significant; Kruskal–Wallis test. Scale bar: 0.2 μm.

F  Cartoon showing that sporozoites cannot form in the absence of microtubules, while lower numbers allowed formation of aberrantly shaped parasites. Note that subpellicular microtubules are exclusively polymerized from the apical polar ring.

Source data are available online for this figure.

---

Taken together, these data show that *α1-tubulin* is the only *α-tubulin* expressed at relevant levels from mid- to late stage oocysts and during sporozoite formation for which it is essential. The data also show that microtubules are not needed for genome segregation and nuclear division in oocysts.

**Aberrant sporozoites are formed by intermediate numbers of microtubules**

As microtubule formation depends on the number and nature of tubulin dimers, we speculated that by modulating gene expression levels or by complementing the *α1-tubulin*(-) parasites with *α2-tubulin* we could generate parasite lines with different microtubule numbers or length. The *α1-* and *α2-tubulin* genes of *Plasmodium* are over 95% identical but show non-synonymous differences in distinct regions of their coding sequence as well as differently sized introns (Fig EV1B and Appendix Fig S4). To investigate whether the sequences of introns or exons play a role in gene expression, we first generated a parasite line with an *α1-tubulin* gene lacking introns (Appendix Fig S5). This resulted in normally shaped sporozoites (Figs 4A and EV5A). In a second parasite line, we additionally codon-modified the exons of *α1-tubulin* (Appendix Fig S5). Such modification did not yield any phenotype in any of the parasite lines we generated so far using the same strategy (Moreau *et al*, 2017; Douglas *et al*, 2018). In contrast, this codon-modified and intron-free *α1-tubulin* line produced highly aberrantly shaped sporozoites. Scanning electron microscopy (SEM) revealed that these parasites featured large blebs or extensions on the surface or even showed never before seen "branching" sporozoite shapes (Figs 4A and EV5A). These malformed sporozoites often showed only faint nuclear staining or two nuclei (Fig 4B and Appendix Fig S6A) and completely failed to enter the salivary gland (Table 2).

We next investigated microtubule length and staining intensity by quantitative fluorescence analysis of SiR-tubulin stained, haemolymph-derived sporozoites. This showed that the malformed sporozoites contained microtubules of the same length as control parasites (Fig 4B). However, quantitative measurements of fluorescence intensity revealed a much lower signal for the sporozoites expressing the codon-modified *α1-tubulin* (Fig 4B). This suggested that fewer microtubules were present in these aberrantly shaped parasites but that these could grow as long as in the wild type. As fewer microtubules might be formed due to lower expression of the

mutated *α1-tubulin,* we performed qRT–PCR analysis. This showed that the expression level of *α1-tubulin* mRNA was lower at day 7 post-infection in both, oocysts expressing "intron-free" and "codon-modified" *α1-tubulin*. However, at day 12, the oocysts expressing intron-free *α1-tubulin* reached the same level of *α1-tubulin* expression as wild types, while in oocysts expressing codon-modified *α1-tubulin*, the expression stayed very low (Fig 4C). This suggested that fewer mRNA would make fewer α1-tubulin protein and hence either shorter or fewer microtubules. To investigate microtubule numbers, we examined oocysts with transmission electron microscopy (TEM). This revealed that the malformed sporozoites had a smaller diameter compared to wild-type parasites as suggested by the SEM images (Fig 4D and Appendix Fig S7) and indeed contained only a median of 6 microtubules (Fig 4E and Appendix Fig S8).

Taking together, these data showed that the introns played only a minor role in modulating *α1-tubulin* expression, but that codon-modification led to lower levels of *α1-tubulin* mRNA and likely α1-tubulin protein. This in turn resulted in fewer microtubules and malformed sporozoites (Fig 4F).

**α2-tubulin can only partially complement α1-tubulin**

α1-tubulin and α2-tubulin differ at several positions including at the C-terminus, where α1-tubulin contains three additional amino acids (Appendix Fig S4). The C-terminus is critical for posttranslational modification and microtubule stability, suggesting that the presence of these three amino acids in α1-tubulin might be important (Gadadhar *et al*, 2017; Magiera *et al*, 2018). To test this as well as the potential role of the other divergent regions, we generated four parasite lines where *α1-tubulin* was replaced by *α-tubulin* constructs partially or fully resembling *α2-tubulin* (Fig 5A and Appendix Fig S5). All lines formed sporozoites (Table 1) but SEM readily revealed some striking differences between the sporozoites of the different lines. The line lacking the last three amino acids showed a similar shape as wild-type sporozoites (Figs 5A and EV5). In contrast, the line termed α2$^{+}$ appeared shorter while the lines termed α2$^{++}$ and α2$^{+++}$ showed aberrantly shaped sporozoites of various length with α2$^{++}$ sporozoites looking less aberrant than those expressing α2$^{+++}$, which often showed rippled surfaces and broader shapes (Fig EV5B).

We next used quantitative fluorescence analysis of SiR-tubulin-staining to compare haemolymph-derived sporozoites of the different

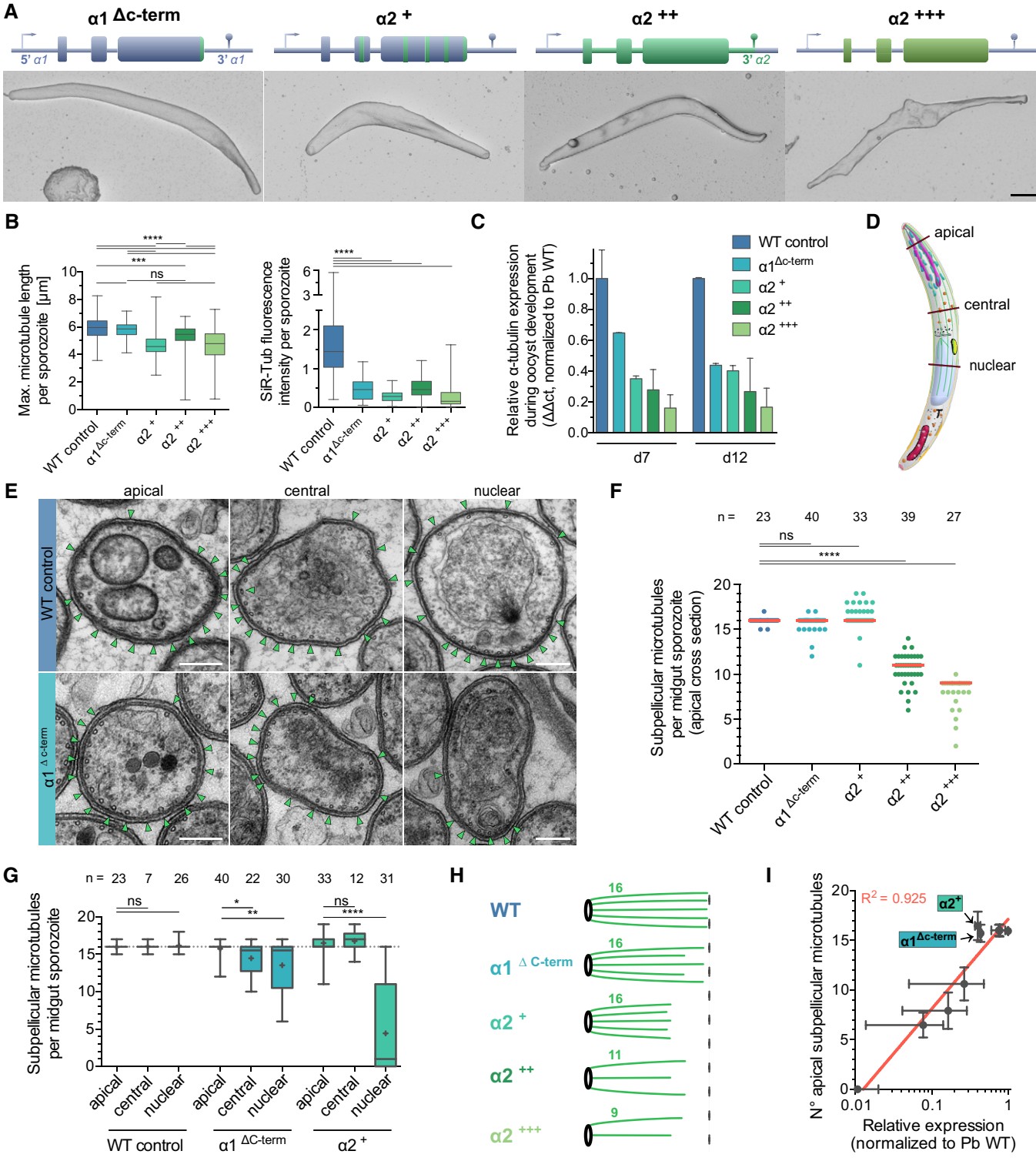

**Figure 5.**

lines. This showed clear differences in microtubule length and staining intensity (Fig 5B and Appendix Fig S6). Even the normal appearing sporozoites from the line lacking the three C-terminal amino acid residues showed a massive reduction in fluorescence intensity of the SiR-tubulin signal, although the maximum length of microtubules was comparable with wild-type sporozoites (Fig 5B). The α2+ line showed the shortest microtubules while the SiR-tubulin intensity was as weak as for the other three lines.

qRT–PCR analysis revealed a difference in expression level between the lines with the α2+++ line showing the weakest

◄

**Figure 5.  α1-tubulin-specific amino acid residues modulate microtubule length while modulating expression level tunes microtubule numbers.**

A   Expression of different α1-α2 chimeric α-tubulins instead of α1-tubulin leads to sporozoite shape changes as revealed by SEM. See also Fig EV5. Scale bar: 1 μm. α1$^{\Delta c\text{-term}}$ lacks the last 3 amino acid residues of α1-tubulin. α2$^+$ parasites contain an α1-tubulin with all amino acid residues that are different between the two α-tubulins changed into those of α2-tubulin; see also Appendix Fig S4. α2$^{++}$ parasites contain *α2-tubulin* (exons, introns, 3′UTR), and α2$^{+++}$ parasites express the exons of *α2-tubulin* and introns of *α1-tubulin* from the *α1-tubulin* promoter.

B   SiR-tubulin staining reveals differences in microtubule lengths and intensity. Between 69 and 156 sporozoites were analysed per line. Box plots represent 50% (boxes) and 95% (bars) of data with horizontal bar showing the median. *** and **** indicate $P < 0.001$ and $P < 0.0001$, respectively; ns: not significant; Kruskal–Wallis test.

C   qRT–PCR analysis of the parasite lines at day 12 after infection shows different expression level. Note the subtly higher expression of α2$^{++}$ compared to α2$^{+++}$. Error bars show standard deviation from the mean of two biological replicates.

D   Schematic cartoon of sporozoite indicating the apical, central and nuclear sections as analysed in panels (E–G).

E   TEM along sporozoites shows a decrease in microtubule numbers towards the nucleus in parasites expressing α-tubulin chimeras. Green arrowheads point to microtubules. See also Appendix Figs S8 and S9. Note that the WT control panel is shown already in Fig 4E to indicate that here we perform an extended analysis to the dataset generated earlier. Scale bars: 0.2 μm.

F   TEM reveals different numbers of microtubules at the apical end of sporozoites for the different lines. Each dot represents the numbers of microtubules counted per section. Red lines indicate median from 23 to 40 analysed sections per line. **** indicates $P < 0.0001$; ns: not significant; Kruskal–Wallis test.

G   Microtubule quantification along the sporozoite reveals the presence of fewer microtubules in central and nuclear sections of α2$^+$ and α1$^{\Delta c\text{-term}}$ parasites compared to wild-type sporozoites. Box plots represent 50% (boxes) and 95% (bars) of data with horizontal bar showing the median and + the mean. *, ** and **** indicate $P < 0.05$, $P < 0.01$ and $P < 0.0001$, respectively; ns: not significant; Kruskal–Wallis test.

H   Cartoon summarizing the effect of α2-tubulin expression on microtubule numbers and lengths. Numbers indicate the median number of observed microtubules per line.

I   Semilog correlation of gene expression as determined by qRT–PCR with microtubule numbers as determine by TEM. Bars show standard deviation around the mean (dots) of two biological replicates (x) and all determined values (y).

Source data are available online for this figure.

expression, and we found little difference between the α2$^+$ line and the line lacking the three C-terminal amino acids (Fig 5C). To investigate whether these sporozoites showed different numbers of microtubules, we turned again to TEM of oocysts, where sporozoites are densely packed. This showed that there were fewer microtubules (a median of 11 and 9, respectively) present at the apical tip in the α2$^{++}$- and α2$^{+++}$-expressing lines. These express *α2-tubulin* (including the introns and 3′UTR) instead of *α1-tubulin* (α2$^{++}$) or just the exons of *α2-tubulin* (with introns and 3′UTR of *α1-tubulin*; α2$^{+++}$; Fig 5D–F and Appendix Fig S9). This suggests that the introns and 3′UTR of the *α2-tubulin* gene might subtly increase gene expression, possibly by affecting RNA stability leading to more tubulin and thus more microtubules.

The discrepancy of microtubule numbers detected by TEM and the weak SiR-tubulin signal suggested that microtubules might be shorter in those parasites. To assess microtubule numbers and length along the sporozoite, we investigated TEM images from oocysts, which could be classified as "apical" if the secretory vesicles important for invasion called rhoptries and micronemes were present, "nuclear" if a nucleus was detected in the section and "central" if neither was clearly present but microtubules were seen below the IMC. Analysis of over 200 sporozoite cross sections from three lines showed that few microtubules reached the nucleus in parasites expressing a C-terminally truncated α1-tubulin (α1$^{\Delta c\text{-term}}$) and in the α2$^+$ line (expressing α2-tubulin as generated by point mutations in the *α1-tubulin* gene; Fig 5E and G). This suggests that microtubule length is determined by the nature of the α-tubulin that is expressed, with α2-tubulin expression leading overall to shorter microtubules (Fig 5H). Most importantly, however, these data suggest a correlation of α-tubulin expression levels with the number of cytoplasmic microtubules, which was supported by regression analysis (Fig 5I).

## A mathematical model predicts that expression level dictates microtubule number

Because an experimental determination of protein levels and assembly dynamics turned out to be impossible in the context of

sporozoites growing in oocysts in infected mosquito midguts (mainly due to contamination from mosquito midgut tubulin), we next turned to mathematical modelling to understand why α-tubulin expression levels mainly affect microtubule numbers but not length. We developed a model for microtubule nucleation and growth from a fixed number of nucleation sites, following earlier work for the dynamics of microtubule length in human cells or reconstitution assays (Kuchnir Fygenson *et al*, 1995; Flyvbjerg *et al*, 1996). We found that the experimental data can be explained very well by a simple two-step nucleation process (Fig 6 and Appendix Supplementary Text). Our analysis shows that at low tubulin concentrations, the growth of formed microtubules decreases available dimers for further nucleation, thereby regulating the number of nuclei that can form at a given concentration. With the increase in concentration, the rate of nucleation increases faster than the microtubule elongation rate; hence, the microtubule number increases. At higher tubulin concentrations, the rate of nucleation is high. As the maximal number of nuclei is limited by the presence of only 16 nucleation sites, the microtubules now grow to full length. For a given tubulin concentration, the rate of binding to nucleation sites and the rate of microtubule growth determine microtubule numbers and length (Fig 6A). To estimate these rates and other relevant model parameters, we compute mean squared error (MSE) in MT number and length between the simulation and experimental results over a wide range of parameter values (Fig 6B and C and Appendix Supplementary Text). Our analysis suggests that there is no optimal estimate, but a trade-off exists between minimum MSE in MT number and MSE in MT length at the corresponding binding rate (shown by open circles in Fig 6B and C) for different values of binding cooperativity to the nucleation site. While the MSE in MT number shows clear minima for not too high values of cooperativity (Fig 6B), the MSE in MT length starts to rise at a threshold in binding rate that depends on cooperativity (Fig 6C). Because our experimental data are very reliable regarding MT number, we first identified the optimal value of binding rate for MSE in MT number and then selected a cooperativity value such that the MSE in MT length is still close to the threshold (choices shown by star symbol in Fig 6B and C). With

the estimated parameters, the model reproduces experimentally observed microtubule number and length at different concentrations (Fig 6D and E). We also plotted the temporal dynamics of average microtubule length and the number of nuclei for wild type and α1$^{\Delta C\text{-}term}$ expression level as obtained from stochastic computer simulations (Fig 6F and G). The model predicts that average microtubule

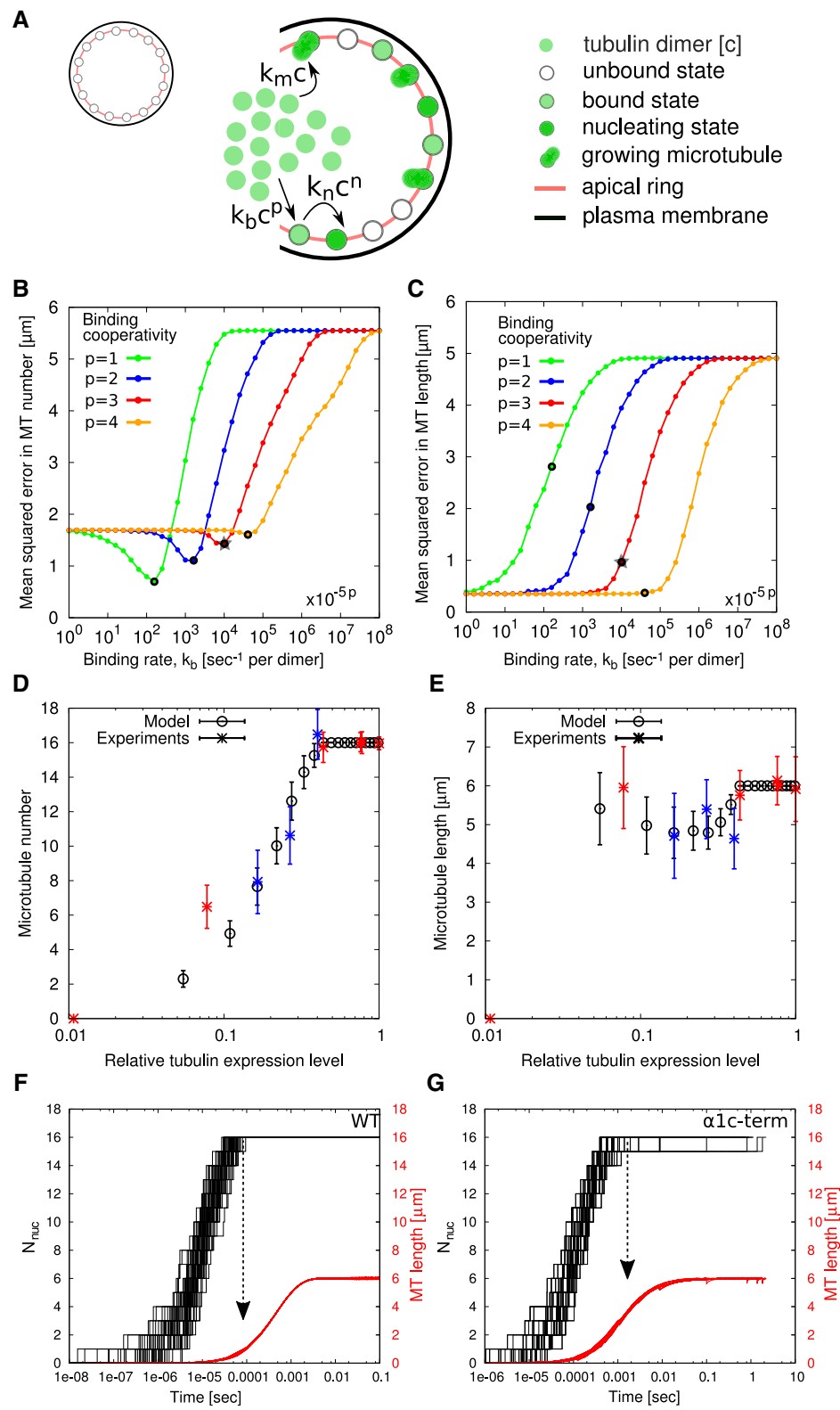

Figure 6.

**Figure 6. Mathematical model explains average number and length of microtubules at different tubulin expression levels.**

A  A cartoon depicting four possible states of a microtubule nucleation site considered in the mathematical model, namely unbound, bound, nucleating and microtubule growing, as well as the transition between them. The transition rates depend on the dimer concentration c. An unbound nucleation site transitions to a bound state when a tubulin dimer binds to it with rate $k_b c^p$. A bound nucleation site starts nucleation with rate $k_n c^n$ and adds tubulin dimer sequentially with rate $k_m c$ to grow the microtubule. Here, $p$ and $n$ (where $p + n = 12$ being the size of the critical nucleus, see Materials and Methods) represent the cooperativity of tubulin dimer binding to a free and a bound nucleation site, respectively.

B  The minimum value of MSE in MT number (shown by black points) increases with the increase in binding cooperativity (in curves from left to right).

C  The MSE in MT length at binding rates corresponding to MSE minimum in MT number (shown by open black circles) decreases with increase in binding cooperativity. The parameter set shown by star symbol is selected for comparing the modelling results with the experiments. The x-values are multiplied by a factor of $(10^{-5})^p$ to present the plots in a common range for easy comparisons. The rate of microtubule growth is fixed to $k_m = 66/s$ per dimer in all the simulations (see Appendix Supplementary Text).

D  The microtubule number increases with the increase in the concentration of tubulin dimers. Microtubule number scales with tubulin concentration at low expression. At high expression, microtubule number is limited by the number of nucleation sites. Shown are mean ± standard deviation from all determined values.

E  The average microtubule length remains close to the maximum length 6 μm. The circles are obtained by averaging results from 200 stochastic simulations of the model equations, and stars are the experimental data (blue stars are for ($\alpha 2^+$, $\alpha 2^{++}$, $\alpha 2^{+++}$) strains). The error bar shows the standard deviation in the data and simulation results.

F  For the WT expression level ($c_0 = 36.6(m_0 + p + n)$), the rate of nucleation is faster than the microtubule growth. Then, the microtubule number is determined by the number of nucleation sites. The average microtubule length is around 1 μm (shown by dashed arrow) when the last nucleus forms.

G  At intermediate tubulin numbers ($c_0 = 16(m_0 + p + n)$), corresponding to the α1c-term mutant, the microtubule formation rate is comparable to nucleation rate. The average microtubule length is around 4 μm (shown by dashed arrow) when the last nucleus forms.

Source data are available online for this figure.

length at time of formation of the 16th microtubule nucleus for wild type is shorter (average length of 1 μm) compared to the α1$^{\Delta C\text{-term}}$ mutant (average length of 4 μm), which also forms 16 microtubules but at a lower expression level (44% of wild type) and contains some shorter microtubules (Figs 5G and 6F and G). Overall, our mathematical model shows that the experimentally observed relation between tubulin expression and microtubule numbers and lengths can arise from the competition between nucleation and growth with plausible parameter values.

**Microtubule length and numbers affect sporozoite curvature and infectivity**

In order to address the functional consequences of this unusual regulation of the microtubule system, we finally investigated the capacity of sporozoites from the different lines for salivary gland colonization and transmission to mice (Table 2). This showed that sporozoites expressing α2$^{+++}$ largely failed to enter the salivary glands, while those expressing α2$^+$ and α2$^{++}$ readily entered salivary glands although at reduced levels compared to wild type (Table 2). Like for sporozoites isolated from the haemolymph, α2$^+$ sporozoites from the salivary gland were the shortest sporozoites, suggesting that sporozoite length is not essential for salivary gland invasion (Fig 7A).

In contrast to haemolymph-derived sporozoites (Fig 4B), the salivary gland-derived sporozoites expressing *α1-tubulin* without introns showed a similar maximum length for microtubules but a weaker fluorescent intensity, suggesting that fewer microtubules grow to the maximum length (Fig 7B). This implies that introns might be important for modulating expression to maintain microtubule length. All other lines showed similar ratios as for hemolymph-derived sporozoites. Sporozoites expressing the α2$^+$ mutation featured the shortest microtubules and were shorter than all other sporozoites (Fig 7A and B). Due to their intrinsic curvature and chirality, sporozoites move in circular counter-clockwise paths on a flat surface (Vanderberg, 1974; Kudryashev *et al*, 2012; Fig 7C). Compared to wild-type sporozoites, those expressing α2$^+$ showed a stronger curvature and moved on tighter paths (Fig 7C). Quantitative analysis of sporozoite migration

revealed that only the α2$^{+++}$ sporozoites showed a motility phenotype: fewer sporozoites were gliding (Fig 7D). Intriguingly, the lines featuring fewer microtubules had a tendency to move more often in apparent clockwise direction (Fig 7E), suggesting that their dorsoventral polarity essential for counter-clockwise movement (Kudryashev *et al*, 2012) is perturbed by the lower number of microtubules.

As α2$^{+++}$ sporozoites barely entered salivary glands and featured on average 9 microtubules (Fig 5F and Table 2), while α2$^{++}$ sporozoites featured on average 11 microtubules and readily colonized the glands, we performed TEM on infected salivary glands to determine the number of microtubules of salivary gland resident α2$^{++}$ sporozoites. As sporozoites are not as densely packed in salivary glands as in oocysts, the observation of large numbers of sections was necessary to yield over 30 sporozoite cross sections of sufficient quality to enable the determination of microtubule numbers. This showed that sporozoites entering the glands featured at least 10 microtubules, suggesting that a threshold of a single additional microtubule determines the capacity for salivary gland entry (Fig 7F). Intriguingly, all investigated species of *Plasmodium* form sporozoites with at least 11 subpellicular microtubules (Garnham *et al*, 1963; Garnham, 1966), suggesting an evolutionary constraint on microtubule numbers to form infective sporozoites.

Sporozoites residing in the salivary gland must be transmitted to the mammalian host to complete the life cycle. Sporozoites are deposited in the skin, where they migrate to find a blood vessel, which they need to enter on their way to the liver (Ménard *et al*, 2013; Douglas *et al*, 2015). To test infectivity, we infected outbred NMRI mice with the different parasite lines either by letting mosquitoes bite or by injecting 10.000 salivary gland-derived sporozoites intravenously (*i.v.*) to circumvent the skin phase of sporozoite migration. Infectivity was determined by taking blood smears from the infected mice on a daily basis to determine how long the parasites take to reach the blood. This gives an indication of their pre-erythrocytic infectivity, i.e. their capacity to cross the skin, enter and leave the blood, develop in and egress from the liver. Transmission of both α2$^+$ and α2$^{++}$ lines by bite-infected fewer mice compared to wild type and the mice infected with the α2$^+$ sporozoites showed an additional delay in the

onset of blood-stage infection (Fig 7G and Table 2). This delay was also observed when sporozoites were injected *i.v.* (Fig 7G and Table 2). These data show that not only the number of microtubules but also their length is critically important for the overall infectivity of sporozoites (Fig 7H).

## Discussion

Due to the highly dynamic nature of microtubules, the biological significance of microtubule number and length cannot be readily determined in most eukaryotic cells. Yet, in some cells, microtubule

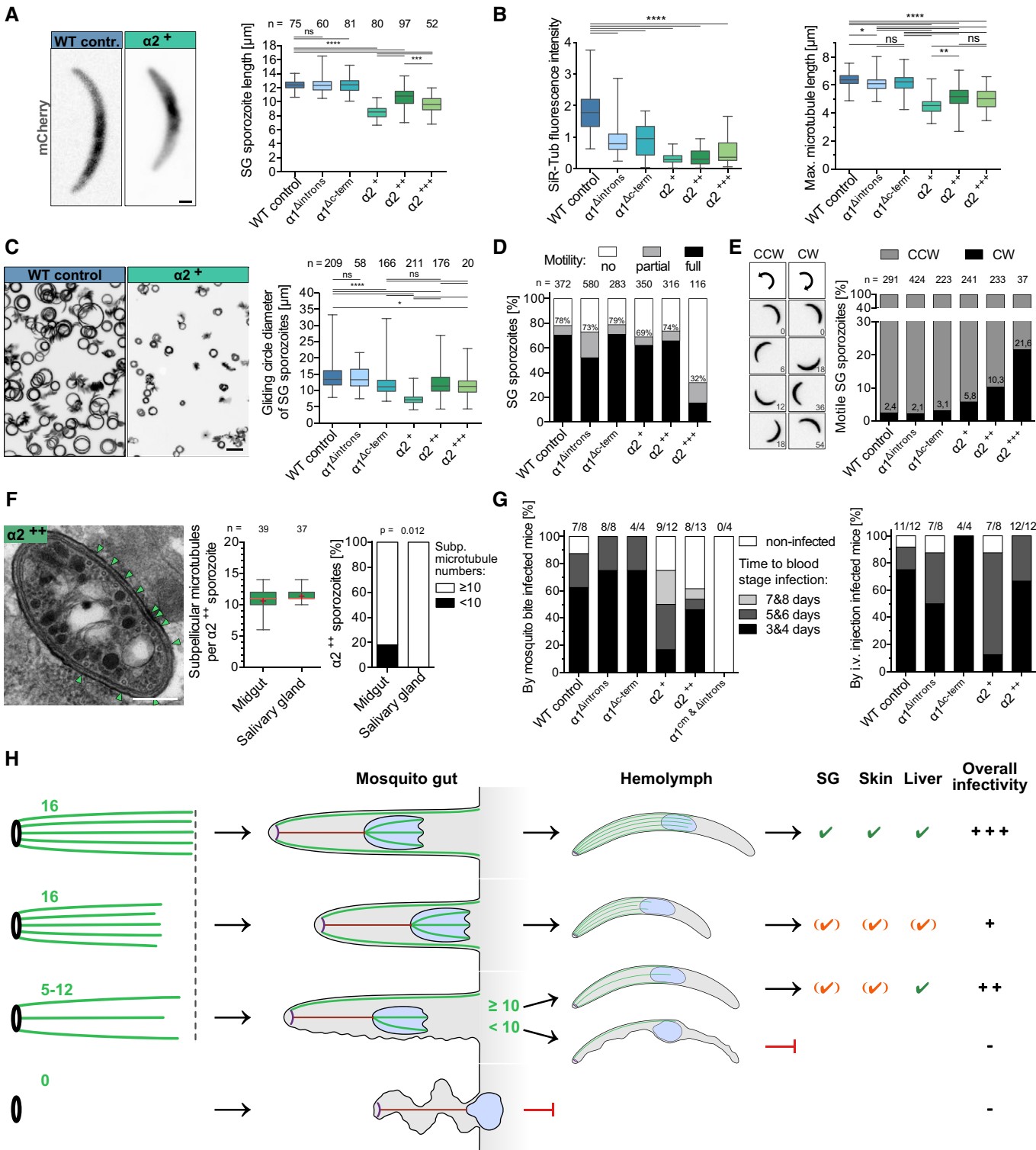

**Figure 7.**

**Figure 7. Microtubule length and numbers determine sporozoite curvature and infectivity.**

A  Quantitative analysis of salivary gland sporozoite lengths from the different lines. *n* indicates numbers of investigated sporozoites. Box plots represent 50% (boxes) and 95% (bars) of data with horizontal bar showing the median. *** and **** indicate $P < 0.001$ and $P < 0.0001$, respectively; ns: not significant; Kruskal–Wallis test; scale bar: 1 μm.

B  Quantification of SiR-tubulin fluorescence from 69 to 127 salivary gland-derived sporozoites. Note the decrease in the fluorescence intensity of sporozoites expressing *α1-tubulin* without introns as well as the decreased intensity and length of the chimeras. Linear correlation of sporozoite length and microtubule length reveals a $R^2 = 0.94$. *, ** and **** indicate $P < 0.05$, $P < 0.01$ and $P < 0.0001$, respectively; ns: not significant; Kruskal–Wallis test.

C  Shorter sporozoites exhibit higher curvatures and hence tighter circular motility on glass. Scale bar: 20 μm. Box plots represent 50% (boxes) and 95% (bars) of data with horizontal bar showing the median. * and **** indicate $P < 0.05$ and $P < 0.0001$, respectively; ns: not significant; Kruskal–Wallis test.

D  Sporozoites with fewest microtubules (α2$^{+++}$) show a motility defect. *n* indicates numbers of investigated sporozoites; numbers on grey bars indicate percentage of partially and persistently moving sporozoites.

E  Sporozoites from lines with fewer microtubules move more frequently in apparent clockwise (CW) direction. Numbers on black bars indicate percentage of CW moving sporozoites; numbers in movie stills indicate time in seconds. CW moving sporozoites move at lower speed than CCW moving ones.

F  Salivary gland sporozoites contain at least 10 microtubules as revealed by TEM of α2$^{++}$-infected salivary glands. Green arrowheads point to microtubules. Scale bar: 0.2 μm. Error bars show the full range and boxes 50% of values. Red crosses indicate the mean, and red lines indicate the median. Fisher's exact test (two-tailed).

G  Infectivity of sporozoites to mice decreases with decreasing numbers of microtubules and shorter microtubules. The prepatency is prolonged in parasites with a reduced sporozoite length and gliding circle diameter. See also Table 2. NMRI mice were exposed to ten mosquito bites of the respective parasite line. Numbers indicate infected mice/mice exposed to mosquito bites (left) and infected mice/mice injected with 10,000 sporozoites (right).

H  Cartoon illustrating the various consequences of different microtubule numbers and lengths during *Plasmodium* sporozoite formation and tissue invasion. Sporozoites with more than 10 microtubules can enter salivary glands while salivary gland sporozoites with shorter microtubules are least infectious.

Source data are available online for this figure.

numbers and length appear tightly regulated as both parameters are conserved over a large number of individual cells. One set of organisms that features different cells where cytoplasmic non-axonemal microtubules appear in fixed numbers and shape are the malaria-causing *Plasmodium spp*. These present extracellular invasive stages during at least three parts of their life cycle with the red blood cell invading merozoites of *P. falciparum* showing 3–4 microtubules (Fowler *et al*, 1998), the mosquito midgut traversing ookinetes showing around 60 (Garnham *et al*, 1962; Morrissette & Sibley, 2002a) and the salivary gland invading, skin traversing and liver infecting sporozoites showing between 11 and 16 microtubules depending on the species (Garnham, 1966). Common to the microtubules in these different forms is their stability, which makes them hard to disrupt using drugs (Morrissette & Sibley, 2002a; Cyrklaff *et al*, 2007).

To address microtubule function in *Plasmodia*, we deleted one of the two *α-tubulin* genes in *P. berghei*, a model malaria parasite infecting rodents. Parasites lacking the *α1-tubulin* gene grew in blood stages formed sexual stages and ookinetes and infected mosquito midguts at rates indistinguishable from wild-type controls. However, they failed to form sporozoites in oocysts. Within oocysts, a diploid genome in a single nucleus replicates into hundreds to thousands of haploid genomes by closed mitosis. Strikingly, this process was not visibly disrupted by the absence of microtubules in the mid-stage to late stage. A similar phenomenon was observed before only in fission yeast, where chromosomes were associated with the spindle pole bodies and actin caused the separation of spindle pole bodies through nuclear ruffling thus distributing the chromosomes to the daughter nuclei (Castagnetti *et al*, 2010). Whether the same process also occurs in oocysts still needs to be investigated but we found evidence of separating centriolar plaques (the equivalents of spindle pole bodies in *Plasmodium*) that lacked microtubules (Fig 3F). In the apicomplexan parasite *T. gondii* kinetochores are also associated with the spindle pole equivalent structure, the centrocone (Farrell & Gubbels, 2014), yet nuclear division could not take place under high enough concentrations of microtubule inhibiting drugs that yielded parasites without microtubules,

while formation of progeny parasites was disrupted (Morrissette & Sibley, 2002b). An indication that actin polymerization could play a role in nuclear division in *Plasmodium* oocysts comes from the deletion of the actin monomer binding and actin filament assembly blocking protein C-CAP that affects nuclear division in oocysts (Hliscs *et al*, 2010). One could speculate that overshooting actin filament formation in the *C-CAP*(-) parasites would allow only the observed 4-5 rounds of nuclear division before it is stalled. A combination of fluorescently marked centriolar plaques and actin filaments could thus inform on the mechanism of nuclear division in the absence of microtubules, although labelling actin filaments in *Plasmodium* is notoriously difficult (Douglas *et al*, 2018). Whatever the molecular mechanism, the absence of a necessity for microtubules in genome separation and nuclear division during sporogony allows evolutionary considerations (Castagnetti *et al*, 2010; Koumandou *et al*, 2013). Could these observations hint to a non-microtubule-based cell division mechanism in the last common eukaryotic ancestor, or are they a sign for a high level of robustness necessary to evolve nuclear division? As for *Plasmodium,* one open key question is whether chromosomes are still reliably separated into the daughter nuclei. The presence of small nuclei in budding sporozoites of our mutants suggests that there might be defects that could be analysed if fluorescence *in situ* hybridization protocols could be developed for oocysts. Furthermore, it is not clear whether chromosome separation in other stages is independent of microtubules.

The stage specific and absolute effect (no microtubules present, no sporozoites formed) of the ablation of *α1-tubulin* allowed us to investigate by a diverse set of genetic complementation experiments, the effect of length and number of microtubules. In parasites showing fewer microtubules but also in wild-type parasites, we noted that the pellicle of the parasite comprising plasma membrane and inner membrane complex was less straight and sometimes undulating at sites where microtubules were absent. This in combination with the shapes found in the *α1-tubulin*(-) parasites strongly suggests that microtubules are necessary to organize the pellicle into a flattened shape that is likely important for motility and invasion.

Generation and analysis of parasite lines that yielded sporozoites featuring a median of 6, 9, 11 or 16 microtubules showed that only sporozoites with 10 or more microtubules could enter salivary glands. This is curious as all investigated parasite species showed sporozoites with at least 11 microtubules, *P. vivax* and *P. cynomolgi bastianelli* being those with the lowest numbers (Garnham *et al*, 1963). This raises the question how different species constrain their sporozoites to the respective number of microtubules that are formed. Are there a maximum number of microtubule nucleation sites present at the polar ring that varies from species to species? Our mathematical model predicted that a lower concentration of tubulin dimers leads to fewer microtubules, but it assumed a constant number of nucleation sites. Hence, also the expression level of tubulin genes might differ from species to species or both, the nucleation site number and expression level have adapted during evolution. Cross complementation of *α1-tubulin* genes from different species might answer this question, but too little is currently known about the regulation of gene expression in *Plasmodium* to make this a sensible experiment.

To our surprise, we found that changing the codon usage of *α1-tubulin* led to a massive reduction in *α1-tubulin* mRNA. Using a similar strategy, we recently also manipulated *P. berghei* actin and found no phenotypic difference (Douglas *et al*, 2018). This raises the question how α-tubulin expression is regulated. Lower mRNA levels could be due to a higher rate of mRNA degradation or a lower rate of mRNA formation. Both could be mediated by RNA binding proteins or non-coding regulatory RNAs, but little is known about either in *Plasmodium*. A comprehensive analysis of RNAs in the blood stages revealed a long non-coding RNA associated with the *α2-tubulin* gene (Broadbent *et al*, 2015) but no data are available for the mosquito stages and hence *α1-tubulin*. Previously, we found that deletion of introns from the gene encoding the actin-binding protein profilin subtly affected expression but without any functional consequence on the parasite (Moreau *et al*, 2017). Similarly, deletion of the introns of *α1-tubulin* led to only a mild reduction in mRNA at day 7 but no detectable difference to wild type at day 12 and hence also no detectable phenotype of the respective parasite line. Yet, the same parasite showed less intense microtubule staining by SiR-tubulin in salivary gland sporozoites and a slightly reduced length (Fig 7B), suggesting that the *α1-tubulin* introns do play a subtle role in gene expression. The subtle but clear effects on *α1-tubulin* expression, microtubule length and number as well as parasite shape and infectivity between the $α2^{+}$-, $α2^{++}$- and $α2^{+++}$-tubulin-expressing sporozoites further suggests that regulation of expression depends on the sequence of both introns and exons.

Another unexpected result from our study was the variable length of the subpellicular microtubules in the mutants expressing *α-tubulin* "chimeras". While the deletion of the last three amino acids in α1-tubulin already reduced the length of some microtubules, the replacement of the other amino acids that are different between α2-tubulin and α1-tubulin ($α2^{+}$-tubulin) reduced the length of microtubules by about 30% and concomitantly the median length of sporozoites from 12.4 to 8.5 μm. These sporozoites were less invasive for salivary glands and also featured a reduced level of infectivity to mice. Whether this reduced infectivity is due to the altered shape or due to a difference in organellar make-up is currently not clear. TEM images of these $α2^{+}$-tubulin sporozoites showed the presence of all key organelles but no quantitative molecular analysis was performed. A difference in shape

could mean a changed dynamic in the skin by, e.g., less spread from the injection site and/or lower probability to associate with blood capillaries (Amino *et al*, 2006; Hopp *et al*, 2015; Muthinja *et al*, 2017). The reduced infection level in mice could result from a problem of the parasites crossing the skin and blood vessels but could also be due to a slower growth in the liver, where a single sporozoite transforms into thousands of progeny merozoites in a process very likely also depending on microtubules.

Another possibility contributing to decreased infectivity of the parasites showing fewer microtubules could be a change in the subpellicular network, a cytoskeletal system underlying the inner membrane complex and being linked to microtubules (Gould *et al*, 2008; Kudryashev *et al*, 2012; Harding & Meissner, 2014; Frisch-knecht & Matuschewski, 2017). It is not clear how this network is established but perturbation by genetic deletion of proteins shows that a destabilization of the network influences sporozoite shape and tensile strength (Khater *et al*, 2004). It is possible that the subpellicular network interconnects with the microtubules during parasite formation and is affected by a perturbation of microtubules. Indeed, the opposite has been shown recently through deletion of the subpellicular network protein G2 in *P. berghei* sporozoites (Tremp *et al*, 2013) and GAPM1a in *T. gondii* (Harding *et al*, 2019).

Of the sporozoites invading the salivary glands also the sporozoites expressing $α2^{++}$-tubulin and $α2^{+++}$-tubulin (the entire *α2-tubulin* gene or the *α2-tubulin* exons and introns of *α1-tubulin*, respectively) showed shorter microtubules and a stronger curvature, if not as dramatic as those expressing $α2^{+}$-tubulin. Their defects in gliding motility provide further evidence that microtubules are important in generating a rigid sporozoite pellicle and a frame against which the actin-myosin-based motor machinery can pull (Heintzelman, 2015). The striking shift towards more frequent "clockwise" gliding of the chiral sporozoites supports a model, where the polarized distribution of microtubules enables the robust motility in only "counter-clockwise" direction (Kudryashev *et al*, 2012). The effect of this shift for *in vivo* migration is currently unclear but could be imaged and quantified in the skin or in 3D environments (Muthinja *et al*, 2018).

Importantly, we found that the difference in expression of *α2-tubulin* instead of *α1-tubulin* resulted in shorter microtubules if enough mRNA and hence tubulin is made. Curiously in about 25% of investigated $α2^{+}$-tubulin sporozoites, we found one or two more microtubules than in control parasites. This might reflect a difference in microtubule assembly dynamics. Indeed, α2-tubulin is likely needed for formation of gametes (Kooij *et al*, 2005) and these need to build up axonemes in minutes compared to the likely several hours needed to build full-length microtubules in budding sporozoites. Ultimately, to address this and other questions raised by our study will necessitate to image sporozoite formation over time, which has to date not been achieved. A necessary breakthrough towards this end would be the reliable cultivation of oocysts *in vitro*. Such cultivation has been described before (Al-Olayan *et al*, 2002) but could not be reproduced robustly in other laboratories (Azevedo *et al*, 2017).

Finally, variations of our approach could help to develop microtubule-targeting drugs, which are available for treatment of cancer and infections by some worms. As we generated parasites expressing α2-tubulin in place of α1-tubulin, we could also envisage to

replace the essential α2-tubulin with α1-tubulin, or if not possible with chimeras of the two α-tubulins that contain just one or some of the subtle differences between the two isoforms. This might yield insights into the essentiality of α2-tubulin in blood and sexual stages. We anticipate α2-tubulin to be specifically important for the formation of the axoneme in male gametes, the only form of the *Plasmodium* parasite with a flagellum. Furthermore, mutations could be introduced into both α2-tubulin and β-tubulin to test the specificity of known microtubule binding drugs. This approach could then be expanded to new microtubule binding molecules. As a proof of concept towards this direction, one could introduce the albendazole- or colchicine-binding residues into *P. berghei* β-tubulin, which should generate drug-sensitive parasite lines (Enos & Coles, 1990; Banerjee *et al*, 2007) or mutate the dinitroaniline binding site in α-tubulin to generate resistant parasites (Lyons-Abbott *et al*, 2010; Ma *et al*, 2010).

Taken together, our data show that the length and numbers of microtubules are essential determinants for generating the *Plasmodium* sporozoite shape and to ensure parasite infectivity (Fig 7H). The importance of precise regulation of microtubule numbers and length for cellular morphology and cell migration as shown here for a medically relevant unicellular eukaryote impacts our understanding of microtubule function and could inform the search for new microtubule-targeting anti-malarials (Kappes & Rohrbach, 2007). Our approach to decipher the role of microtubule numbers and lengths by perturbing tubulin expression can also readily be applied to other model systems of broad biological relevance.

# Materials and Methods

## Bioinformatic analysis

Sequences were retrieved from PlasmoDB (version 36; Aurrecoechea *et al*, 2009) and UniProt (2018; Bateman *et al*, 2017). Sequence alignments were conducted with CLC Main Workbench 8 (Qiagen) or SnapGene (version 3.2.1).

## Oligonucleotide primers and enzymes

Unless otherwise noted, we purchased primers from Thermo Fisher Scientific (for sequences, see Appendix Table S1) and enzymes from New England Biolabs.

## Construction of plasmids

Plasmids were designed for double-homologous crossover integration into the *Plasmodium* genome. The Pb262 vector (Singer *et al*, 2015) was modified by replacing the chromosome 12 integration sites with *α1-tubulin* 5′UTR and 3′UTR integration sites for all vectors used in this study. Furthermore, the *mCherry* open reading frame (ORF) and the *dhfs* 3′UTR were replaced with a set of different versions of *α1-tubulin* or *α1/α2-tubulin* chimeras and the *α1-tubulin* 3′UTR, respectively (Fig EV2A and Appendix Fig S5). In the case of the α2$^{+++}$ vector, the *dhfs* 3′UTR was replaced by a *α2-tubulin* 3′UTR (Appendix Fig S5). In case of the *α1-tubulin*(-) vector, the *mCherry* and the *dhfs* 3′UTR were deleted (Fig EV1).

Amplification of cDNA or genomic DNA was performed by polymerase chain reaction (PCR) using Phusion® High-Fidelity DNA Polymerase (NEB), and insertion/fusion of fragments was performed in a single step reaction by NEBuilder® HiFi DNA Assembly (NEB). For primers, see Appendix Table S1. All vectors contained the positive–negative selection marker *hdhfr-yfcu* and could be used for either the "gene insertion/marker out" technique (Lin *et al*, 2011) or the standard transfection protocols (Janse *et al*, 2006a). Final vectors were linearized before transfection by SalI and KpnI in case of the GIMO technique or by SalI and XhoI (or ScaI) for standard transfection.

## Generation of parasite lines

The linearized *α1-tubulin*(-) vector was transfected into an unmodified *P. berghei* strain ANKA or *P. berghei* strain ANKA expressing mCherry under the CSP and eGFP under the ef1α promoter (Klug & Frischknecht, 2017) using standard protocols (Janse *et al*, 2006a,b). Parasites that integrated the desired DNA construct were selected by administration of pyrimethamine (0.07 mg/ml) via the mouse drinking water. An isogenic population was obtained by a dilution series, which was then followed by elimination of the positive–negative selection marker *hdhfr-yfcu* by applying 5-fluorocytosine (1 mg/ml; Lin *et al*, 2011; Fig EV1). Another dilution series was performed to obtain an isogenic population of the selection marker-free *α1-tubulin*(-) RG parasites, which were used for complementation approaches with WT (α1$^{α1(-)\ compl}$, Fig EV2), deletion of *α1-tubulin* introns (α1$^{Δintrons}$, Appendix Fig S5) and a set of *α1/α2-tubulin* chimera constructs (α2$^{+++}$, α2$^{++}$, α2$^{+}$; Appendix Fig S5). The *α1-tubulin* codon-modified and intron-deleted construct (α1$^{cm\&Δintrons}$) was transfected into the *P. berghei* strain ANKA. A receiver parasite line only differing in the *dhfs* 3′UTR was used to generate the C-terminally truncated *α1-tubulin* parasite line (α1$^{Δc\text{-}term}$; Appendix Fig S5C) via a "gene insertion/marker out" approach (Lin *et al*, 2011).

To test for correct construct integration, blood-stage parasites were obtained from anesthetized mice (87.5 mg/kg ketamine and 12.5 mg/kg xylazine, Sigma-Aldrich) via cardiac puncture once parasitemia reached 1–2%. The infection rate was monitored by Giemsa (Merck)-stained blood smears. Erythrocytes were lysed in 15 ml cold phosphate-buffered saline (PBS) containing 0.03% saponin. After centrifugation and washing, the genomic DNA was isolated using the Blood and Tissue kit (Qiagen Ltd) according to the manufacturer's protocol. All generated isogenic parasite lines were analysed via PCR (Figs EV1 and EV2 and Appendix Fig S5C) and Sanger sequencing (GATC) for error-free construct integration. Parasite lines with the correct genotype were assumed to be isogenic.

## Generation of isogenic parasite lines

Parasites from transfections reflected a mixed population of parasites. A dilution series was performed by injecting only a single blood-stage parasite into each of 5–10 NMRI mice. Once mice reached between 1 and 3% parasitemia, blood was collected with whole blood aliquots stored in liquid nitrogen and the remaining material was used to isolate genomic DNA, which was tested for correct construct integration.

## Mosquito infection

Frozen parasite stocks were injected intraperitoneally (100–150 μl) into 2–3 mice, and parasites were allowed to grow for 4–6 days. Infection rate was monitored by blood smears. When infected mice reached 2–3% parasitemia, mice were anesthetized and fed to *Anopheles stephensi* (Strain Sda 500) mosquitoes.

## Analysis of oocyst development

Midguts of 10–20 mosquitoes were isolated in cold PBS at days 5, 7, 10, 12, 14 and 19 post-mosquito blood meal. Non-fluorescent oocysts were fixed with NP40, stained with 0.1% mercurochrome (Usui *et al*, 2011) and counted by light microscopy (Axiovert 200M, Zeiss) using a 10× objective (NA = 0.5, air). Fluorescent oocysts (eGFP/mCherry) were counted using a stereomicroscope (SMZ1000, Nikon). To visualize microtubules, highly infected midguts were incubated in 100 μl RPMI (supplemented with 50,000 units/l penicillin and 50 mg/l streptomycin) with 3 μM SiR-tubulin (Spirochrome) and 3 μM Hoechst 33342 (Thermo Fisher Scientific) for 30 min at 37°C. Midguts and a minimal volume of medium were then transferred with a Pasteur pipette onto a microscopy slide, covered with a cover slip and sealed with paraffin. Samples were observed using a Nikon TE 2000-E microscope equipped with an UltraView ERS spinning disc confocal unit (Perkin-Elmer) with 20× (NA = 0.85, oil) and 60x (NA = 1.49, oil) objectives. 3D reconstructions of oocysts were rendered from *z*-stack images (*z*-distance between 0.5 μm and 1.5 μm) with the 3D Opacity tool of Volocity 6.3 (Perkin-Elmer).

## Sporozoite isolation and counting

Midgut (MG), hemolymph (HL) and salivary gland (SG) sporozoites of infected mosquitoes were isolated on day 14 and day 17/18 post-mosquito blood meal from at least two independent mosquito feeds. A minimum of 10 mosquitoes was dissected per count. HL sporozoites were isolated from immobilized mosquitoes (on ice) by cutting of the last segment of the abdomen, flushing the abdomen by injecting RPMI into the thorax and collecting drops containing the HL sporozoites from the abdomen. For MG and SG sporozoites, the respective organs were dissected on ice and crushed before counting. Sporozoite numbers were counted using a Neubauer counting chamber. Sporozoites of the different compartments were collected from the same set of mosquitoes to ensure a correct salivary gland invasion ratio calculation.

## Sporozoite movement, length and gliding diameter analysis

HL and extracted salivary glands were collected in 50 μl RPMI on ice between days 17 and 19 post-infection. SGs were crushed with a pestle and centrifuged for 3 min at 1,000 rpm/100 g (Thermo Fisher Scientific, Biofuge primo) to remove salivary gland fragments, and the supernatant was collected. HL sporozoites were concentrated by centrifugation for 3 min at 10,000 rpm/10,000 g. An equal volume of RPMI containing 6% bovine serum albumin (ROTH) was added to the HL and SG sporozoites, and the resulting mixtures were transferred into an optical bottom 96-well plate (Thermo Fisher Scientific). The plate was centrifuged for 3 min at 1,000 rpm/200 g (Multifuge S1-R, Heraeus) and imaged using an epifluorescence microscope (Axiovert 200M, Zeiss). Movies were taken with either differential interference contrast (DIC) or in the mCherry channel using a 10× (NA 0.5, air) or 25× (NA 0.8, water) objective at a speed of 1 frame every 3 s. Videos were analysed for 100 s with Fiji (Version: 2.0.0 rc 64/1.51s; Schindelin *et al*, 2012). Gliding motility was categorized into three different patterns. Fully (persistently) moving sporozoites moved at least one full circle during 100 s. Partially moving sporozoites moved for at least a sporozoite length, and non-moving sporozoites were not moving at all. Non-moving also included attached, waving, twitching, patch gliding and floating sporozoites (Hegge *et al*, 2009). Moving sporozoites were further categorized into clockwise (CW) movers when sporozoites moved for at least two frames in CW direction during the 100-s video. To assess the gliding diameter of sporozoites, 100-s videos were combined by maximum intensity projection of the fluorescent channel and the diameter of the gliding circles was measured. Sporozoite length was measured with the segmented line tool of Fiji (Schindelin *et al*, 2012).

## Microtubule staining assay

Sporozoites were isolated on day 14 post-mosquito blood meal and purified as described above. Sporozoites were mixed in a 1:1 ratio with RPMI containing 6% bovine serum albumin (ROTH), 0.5 μM SiR-Tub and 10 μg/ml Hoechst. Sporozoites were incubated for 10 min at room temperature and then diluted in a 1:5 ratio with RPMI. Sporozoites were then centrifuged onto cover slips at 1,500 rpm/450 g (Multifuge S1-R, Heraeus) for 5 min. Cells were directly fixed in 4% paraformaldehyde (PFA) solution (diluted in PBS) for 5 min and then diluted by 1:4 in PBS and incubated for further 15 min. Cover slips were shortly air-dried and mounted to a microscopy slide with 5 μl ProLong Gold anti-fade reagent (Invitrogen). After 24 h at room temperature, images of sporozoites were acquired with a Nikon TE 2000-E microscope equipped with an UltraView ERS spinning disc confocal unit (Perkin-Elmer) using a 60× objective (NA 1.49, oil). For analysis of blood-stage parasites, 100 μl of whole blood was washed in 1 ml pre-warmed sterile PBS and allowed to settle on concanavalin A (Sigma)-coated chambered cover slips (Lab-Tek). After gentle washing with PBS to remove unbound cells, remaining cells were stained with imaging medium (20% FBS, 25 mM Hepes, 3 μg/ml gentamicin in RPMI-1640 butter, no phenol red) containing 1 μM SiR-Tub for 3–4 h at 37°C. Hoechst 33342 was added to a final concentration of 1 μg/ml shortly before imaging as described above.

## Quantification of microtubule length and intensity

Sporozoite images acquired from the fluorescence assay were analysed with Volocity Analysis 6.3 (Perkin-Elmer). First, a sporozoite was automatically identified by the cytoplasmic mCherry signal and the Hoechst staining of the nucleus. The dynamic range of the original SiR-tubulin staining was adjusted to the weakest microtubule staining near the nucleus. The program quantifies original intensity levels of the SiR-tubulin stain and maximal microtubule length by measuring the longest axis. Automatic measurements were compared to manual measurements obtained with ImageJ (Appendix Fig S1).

## Transmission electron microscopy (TEM)

Blood stages or highly infected MGs or SGs were fixed in 2% paraformaldehyde and 2% glutaraldehyde diluted in 100 mM

sodium cacodylate buffer at 4°C overnight. Fixed samples were washed three times in 100 mM sodium cacodylate buffer at room temperature (RT) for 5 min. A secondary fixation was performed in 1% osmium tetroxide (in 100 mM sodium cacodylate buffer) at RT for 60 min. Samples were washed twice with 100 mM sodium cacodylate buffer and twice in $ddH_2O$ and then contrasted with 1% uranyl acetate (in $ddH_2O$) at 4°C overnight. Samples were washed twice with $ddH_2O$ for 10 min and then dehydrated by incubating in increasing concentrations of acetone (30, 50, 70, 90%) for 10 min and two times in 100% for 10 min. Samples were adapted to "Spurr" solution (23.6% epoxycyclohexylmethyl-3,4-epoxycyclohexylcarboxylate (ERL); 14.2% ERL-4206 plasticizer; 61.3% nonenyl succinic anhydride; 0.9% dimethylethanolamine) by incubating in increasing concentrations (25, 50, 75%) at RT for 45 min and at 100% at 4°C overnight. MGs were resin embedded with "Spurr" at 60°C overnight. Embedded MGs were trimmed, and 70-nm thick sections were imaged on a transmission electron microscope at 80 kV (JEOL JEM-1400) using a TempCam F416 camera (Tietz Video and Image Processing Systems GmbH, Gauting).

### Scanning electron microscopy (SEM)

Sporozoites were isolated as above and fixed in 2% glutaraldehyde and 4% PFA onto cover slips at RT for 1 h or at 4°C overnight. Sporozoites were dehydrated with increasing ethanol concentrations (30, 50, 70, 90% in water) at room temperature for 10 min and in 100% two times for 10 min. Ethanol was exchanged first by incubating the sample with 50% HMDS (hexamethyldisilazane) and 50% ethanol for 5 min and then by 100% HDMS for 10 min. The sample was kept under the fume hood until all HDMS was evaporated. Cover slips were mounted onto studs and sputter-coated with 5–10 nm gold. Sporozoites were imaged using a scanning electron microscope (Leo 1530, Zeiss).

### Mosquito midgut-derived sporozoite tomography

The tomograms were acquired from serial sections of resin-embedded (Spurr) mosquito midguts. Each section (100 and 300 nm thickness for WT and *α1-tubulin*(-) parasites, respectively) was inspected for suitable objects and mapped using a JEOL JEM-1400 80 kV TEM. The tilt series were performed on a FEI Tecnai F30 300 kV TEM with the Gatan OneView sensor (Gatan Inc, Pleasanton, CA, USA) installed and controlled by SerialEM (Mastronarde, 2005). Each series ranged from ± 60–70° with images at 2° increments at 9,600× magnification. The tomogram volumetric reconstruction for each individual section was performed using the IMOD 4.9 software package (Kremer *et al*, 1996). Every image in the tilt series was aligned and tracked via patch tracking, and the volume was reconstructed using weighted-back projection. The 3D reconstructions of the sections were flattened and trimmed before combining them into a single volume. For visual representation, the objects of interest in each tomogram were manually segmented in 3dmod (IMOD). The animations (Movies EV1 and EV2) were created by exporting frames in 3dmod and combining them using FFmpeg (FFmpeg Developers, version: be1d324).

### Expression levels of *α1-tubulin* and *α2-tubulin*

Total RNA of 17 well-infected mosquito midguts (> 1 million sporozoites) was isolated on days 5, 7, 10, 12 and 14 post-infection for WT and *α1-tubulin*(-) parasites and from days 7 and 12 for all other parasite lines. RNA was isolated with Qiazol reagent according to the manufacturer's protocol (Invitrogen). RNA was treated with the Turbo DNA-free kit (Life Technologies) according to the manufacturer's protocol, and cDNA synthesis was generated using the First-Strand cDNA synthesis kit (Thermo Fisher Scientific) and checked afterwards for gDNA contaminations via RT–PCR. The quantitative PCR was performed using SYBR Green PCR Master Mix (Life Technologies) including ROX dye and was measured with the Abi 7500 Fast RT–PCR system (Applied Biosystems). The 18S rRNA gene was used as a reference, and across-run differences were normalized using a calibrator sample. Relative copy numbers were calculated by applying the ΔΔCt methodology. The sequences of the gene-specific primers used are shown in Appendix Table S1.

### Mouse infection by mosquito bites or intravenous sporozoite injection

We tested the ability of vector-to-host transmission of the generated parasite strains by either infecting naïve mice with infected mosquitoes or injecting sporozoites intravenously. Ten preselected infected mosquitoes were put into cups and starved overnight. NMRI mice were anesthetized with ketamine and xylazine (87.5 mg/kg ketamine and 12.5 mg/kg xylazine), and one mouse was put on each cup. The eyes of the mice were covered with Bepanthen cream (Bayer) to prevent dehydration. Mosquitoes were allowed to bite for 20 min. For intravenous injections, SG sporozoites were isolated in RPMI as described above, diluted to 10,000 sporozoites per 100 μl and the same volume was injected into the tail vein per mouse. Parasitemia of mice was monitored daily from days 3 to 8 post-mosquito blood meal or post i.v. injection via blood smears stained with Giemsa solution (Merck). Blood smears were counted via a light microscope (Zeiss) counting grid. The time until the first parasite was identified is stated as the prepatency.

### Parasite blood-stage growth

Blood-stage growth was assessed by injecting intravenously 100 blood-stage parasites per C57Bl/6 mouse. The injection volume was 100 μl. Mice were monitored from days 3 to 10 post-injection as described previously. Growth rate was calculated for day 7, when parasitemia was between 0.7 and 1.5%.

$$\text{growth rate} = \left( \frac{\text{number of parasites}}{\text{number of injected parasites}} \right)^{\frac{1}{\text{day}}}$$

### Parasite strain

All genetic modifications were performed in *Plasmodium berghei* strain ANKA wild-type or in wild-type-derived parasites.

## Ethics statement

Animal experiments were approved by the German authorities (Regierungspräsidium Karlsruhe, Germany) and were performed according to the FELASA and GV-SOLAS standard guidelines. All experiments, with the exception of blood-stage growth assays, were conducted in female NMRI mice (6–8 weeks of age) obtained from JANVIER. Parasite blood-stage growth rates were determined with female C57Bl/6 mice (6–8 weeks of age) obtained from Charles River Laboratories.

## Statistics

Statistical significance was assessed using GraphPad Prism 6.0 h and a one-way ANOVA test (Kruskal–Wallis test).

**Expanded View** for this article is available online.

## Acknowledgements

We thank M. Singer, C. Moreau, P. Rothhaar, R. Douglas, C. Sommerauer and U. Amelung for help with experiments and M. Reinig for mosquito rearing; S. Gold, C. Funaya and S. Hillmer from the Electron Microscopy Core Facility of Heidelberg University as well as M. Schorb and Y. Schwab from the EM core facility of the European Molecular Biology Laboratory, Heidelberg for help with EM; R. Beyeler, S. Castagnetti, R. Douglas, R. Dzikowski, O. Fackler, V. T. Heussler, M. Ganter, J. Guizetti, A. Hakimi, M. Marti and M. Meissner for discussions. We acknowledge the microscopy support from the Infectious Diseases Imaging Platform (IDIP) at the Center for Integrative Infectious Disease Research. This work was funded by fellowships from the EU FP7 network of excellence EVIMalaR and the Hartmut-Hoffmann-Berling International Graduate School of Heidelberg University (BS), the CellNetworks Cluster of Excellence at Heidelberg University (PP), the Deutsche Forschungsgemeinschaft (DFG, German Research Foundation)—Projektnummer 240245660—SFB 1129 (USS, FF), Human Frontier Science Program Young Investigator grant RGY066 (FF) and the European Research Council (StG 281719) (FF).

## Author contributions

BS and FF designed the experiments and wrote the manuscript. BS, HF, PK, CDB, MB performed the experiments. BS analysed data, performed 3D SD modelling and generated the figures. PK performed 3D reconstructions from TEM serial sections with guidance of MC and BS; PP and USS developed the mathematical model. Correspondence and requests for materials should be addressed to FF (freddy.frischknecht@med.uni-heidelberg.de).

## Conflict of interest

The authors declare that they have no conflict of interest.

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
