## [Review Process File · The EMBO Journal]

Microtubule number and length determine cellular shape and function in *Plasmodium*

Benjamin Spreng, Hannah Fleckenstein, Patrick Kübler, Claudia Di Biagio, Madlen Benz, Pintu Patra, Ulrich S. Schwarz, Marek Cyrklaff and Friedrich Frischknecht

Review timeline:	Submission date:	22nd Oct 2018
	Editorial Decision:	26th Nov 2018
	Revision received:	28th Mar 2019
	Editorial Decision:	12th Apr 2019
	Accepted:	26th Apr 2019

Editor: Deniz Senyilmaz Tiebe

Transaction Report:

1st Editorial Decision

26th Nov 2018

Thank you for submitting your manuscript for consideration by the EMBO Journal. It has now been seen by three referees whose comments are shown below. As you can see, all referees express interest in your study investigating how microtubule number and length regulate cellular shape and function in *Plasmodium*. However, they also raise concerns that need to be addressed in full before we can consider publication of the manuscript here.

Given the referees' positive recommendations, I would like to invite you to submit a revised version of the manuscript, addressing the comments of all three reviewers. I should add that it is EMBO Journal policy to allow only a single round of revision, and acceptance of your manuscript will therefore depend on the completeness of your responses in this revised version.

REFeree REPORTS:

Referee #1:

In this manuscript, Spreng, Frischknecht and colleagues studied the role of microtubules in the formation and infectivity of sporozoites, which correspond to the form adopted by *Plasmodium* as they travel between the oocytes and the salivary gland of the mosquito. The deletion of the gene encoding for alpha1-tubulin induced severe morphological defects during sporozoites budding and prevented the infection of the midgut and salivary gland of the mosquito. The authors performed an extensive study of the microtubule network architecture based on both fluorescence imaging and electronic microscopy. Both techniques have intrinsic limitations and their associations gave strong support to all authors observations and conclusions. Microtubules appeared essential for bud formation on the sporoblast and the displacement of the nucleus in it. The absence of microtubules led to the formation of small sporozoites with highly convoluted shapes. Surprisingly nuclear divisions could proceed in the sporoblast, in the absence of microtubules. Intermediate levels of alpha1-tubulin deletion, or complete deletion of a1-tubulin compensated by

the expression of full or truncated form of α 2-tubulin, were used to characterize the specific morphologies and intra-cellular architectures of sporozoites containing 1 to 16 microtubules. Interestingly, the various approaches concur with the identification of a sharp threshold of 11 microtubules as the lower possible values for the proper budding of the sporozoite. Surprisingly all microtubules had similar length whatever their number. This observation contrasts with most classical conditions in which microtubule dynamics impose a broad distribution of length. This unusual architecture could be explained by a model based on a difficult nucleation step followed by an easy polymerization. Considering that microtubules apparently have to apply well balanced pushing forces on the sporoblast membrane to extrude properly the future sporozoite, this tight length regulation is likely central for the budding process.

This is overall an excellent study based on solid observations and thoughtful analyses which led to original conclusions about the role of microtubule in the regulation of parasite morphology. I enjoyed very much reading it and found it thought provoking. I have only one concern relative to the model: only the « correct » set of parameters, capable to reproduce the observed length and numbers of microtubules, is shown. It would have been more informative to show the analysis of the entire range of ratio between the efficiency of elongation to nucleation that allows this conformation to be established. In addition, why is it necessary to distinguish the « bound » and the « nucleating » state? Is it because elongation can only occur when all protofilaments have been initiated? Couldn't a simpler model, taking into account nucleation rate and elongate rate only, be capable to reproduce the observed microtubule patterns?

Minor issue

- Although unexplained, it is known that all cells are not equally permeant to SiR-tubulin. I am wondering whether the non-tensed shape of the mutants could prevent the entry of SiR-tubulin at specific stages of sporozoites formation.

- could microtubules formed with α 2-tubulin be more dynamic (and thus compatible with mitosis) than those formed with α 1-tubulin?

Referee #2:

This is an elegant study from the Frischknecht lab that offers fundamental new insight into the role of microtubules in malaria sporozoite formation and infectivity, with implications for cell biology more generally. The study is elaborate and technically challenging, but well conducted and written. The results support the conclusions and are convincing. I have no hesitation in recommending that the paper be accepted for publication.

The authors have not mentioned an important structure of the pellicle, the subpellicular network, that links the subpellicular microtubules and inner membrane complex. Alveolin and other protein components of the SPN contribute to sporozoite shape, size, tensile strength, motility and infectivity, and can affect the distribution of the microtubules. Changing the size and number of microtubules might impact on SPN formation or function, and thus I feel it would be appropriate that a section is devoted to this in the Discussion.

Referee #3:

Plasmodium species encode two α -tubulins (-1 and 2). PlasmoDB data indicate that at least in *P. falciparum*, they are both expressed at all stages of development, with particularly high expression in oocysts.

This manuscript by Spreng et al describes an analysis of the consequences of deletion of the α -tubulin genes. The authors find that α -tubulin-2 is essential, as reported previously (Kooij et al 2005). The authors are successful in their attempt to delete α -tubulin-1. Previous attempts to knock

out the gene for α -tubulin-1 in *P. berghei* have been unsuccessful (Kooij et al 2005). The authors should comment on why they have had success where previous attempts have failed. The authors found that the α -tubulin-1(-) parasites showed no defect in blood stage development nor in the initial stages of mosquito infection. They report that in the α -tubulin-1(-) parasites, microtubules are not present in sporozoites developing within the oocysts and that sporozoites do not develop correctly.

It is not clear what is happening with α -tubulin-2 in these parasites. Previous work shows that alpha-tubulin-2 is transcribed in the asexual blood stages, gametocytes, ookinetes and oocysts (Kooij et al 2005). The authors undertook qRT-PCR analysis of α -tubulin-2 in oocysts and found a low level of expression, which was unchanged in the α -tubulin-1(-) parasites.

Which α -tubulin isoforms are expressed in asexual blood stages and in gametocytes? Is α -tubulin-2 expressed? Does it perform the tubulin functions in these stages? Why is it not observed by SiR-tubulin in early stage the α -tubulin-1(-) oocysts where it is expressed? Can it be confirmed that alpha-tubulin-2 binds SiR-tubulin (cf yeast tubulin which does not).

Based on their studies with the early stages of nuclear division in oocysts the authors conclude that the chromosome segregation needed for nuclear division occurs even in the absence of microtubules. What is driving chromosomal segregation during asexual division? The authors need to present Western and qRT-PCR analysis of alpha and beta tubulins in early in the early oocyst stage as well as SiR-tubulin labelling and EM data in the asexual stage before it can be concluded that chromosome segregation can occur in the absence of microtubules. It appears more likely that α -tubulin-2 performs the various functions of microtubules at all stages of development until the later stages of nuclear division in oocysts, where the levels become insufficient. A careful analysis of α -tubulin-1 and -2 at the protein level would be needed to determine whether α -tubulin-2 is contributing to the division process.

The authors examined the effect of replacing wildtype α -tubulin-1 with a codon-modified α -tubulin-1 that is expressed at a lower level. Interestingly, they found that the lower level of expression was associated with the formation of fewer microtubules that exhibit a similar length to the MTs in wildtype parasites. Similarly, other modifications to α -tubulin-1, such as adding the C-terminal residues from α -tubulin-2 resulting in lower levels of expression and fewer (and in some cases shorter) microtubules. The defects in MT production led to defects in ability of transfer to the salivary glands and to transmit to recipient mice. It is perhaps not surprising that alterations in the levels of expression of tubulins leads to aberrant biology.

The authors developed a mathematical model that supports the conclusion that the concentration of α -tubulin protomers severely limits the rate of nucleation leading to fewer MTs that have a similar length.

In summary, this is a very interesting study but additional work is needed to support the authors conclusions

1st Revision - authors' response

19th Feb 19

Referee #1:

In this manuscript, Spreng, Frischknecht and colleagues studied the role of microtubules in the formation and infectivity of sporozoites, which correspond to the form adopted by *Plasmodium* as they travel between the oocytes and the salivary gland of the mosquito.

The deletion of the gene encoding for alpha1-tubulin induced severe morphological defects during sporozoites budding and prevented the infection of the midgut and salivary gland of the mosquito. The authors performed an extensive study of the microtubule network architecture based on both fluorescence imaging and electronic microscopy. Both techniques have intrinsic limitations and their associations gave string support to all authors observations and conclusions. Microtubules appeared essential for bud formation on the sporoblast and the displacement of the nucleus in it. The absence of microtubules led to the formation of small sporozoites with highly covoluted shapes. Surprisingly nuclear divisions could proceed in the sporoblast, in the absence of microtubules.

Intermediate levels of alpha1-tubulin deletion, or complete deletion of a1-tubulin compensated by the expression of full or truncated form of a2-tubulin, were used to characterize the specific morphologies and intra-cellular architectures of sporozoites containing 1 to 16 microtubules. Interestingly, the various approaches concur with the identification of a sharp threshold of 11 microtubules as the lower possible values for the proper budding of the sporozoite. Surprisingly all microtubules had similar length whatever their number. This observation contrasts with most

classical conditions in which microtubule dynamics impose a broad distribution of length. This unusual architecture could be explained by a model based on a difficult nucleation step followed by an easy polymerization. Considering that microtubules apparently have to apply well balanced pushing forces on the sporoblast membrane to extrude properly the future sporozoite, this tight length regulation is likely central for the budding process.

This is overall an excellent study based on solid observations and thoughtful analyses which led to original conclusions about the role of microtubule in the regulation of parasite morphology. I enjoyed very much reading it and found it thought provoking. I have only one concern relative to the model: only the « correct » set of parameters, capable to reproduce the observed length and numbers of microtubules, is shown. It would have been more informative to show the analysis of the entire range of ratio between the efficiency of elongation to nucleation that allows this conformation to be established.

A: Thanks for the enthusiastic support and interesting questions on the model. Our choice of model parameters is the result of an optimization process that previously has been described in the supplementary subsection "Estimation of model parameters". We agree with the referee that it is appropriate to explain this in more detail in the main text and we, therefore, have added subpanels B and C to Figure 6 that includes a set of additional "non-optimal" parameters. The accompanying text can be found in lines 330 to 341 and is highlighted in yellow.

This analysis shows the variation of mean squared error in MT number and length between the model outcome and experimental result for a wide range of binding rates and binding cooperativity to nucleation site (with a fixed value of the microtubule elongation rate taken from the literature, supplementary text ref. 1). The final set of parameters are chosen such that the MSE is MT number is minimal for a given cooperativity value. The cooperativity value is chosen such that the MSE in MT length is considerably low is close to the threshold in binding rate where it starts to rise.

In addition, why is it necessary to distinguish the « bound » and the « nucleating » state? Is it because elongation can only occur when all protofilaments have been initiated? Couldn't a simpler model, taking into account nucleation rate and elongate rate only, be capable to reproduce the observed microtubule patterns?

A: We make the nucleation a two-step process (naming them as binding to nucleation site and nucleation step) to connect our model to existing models from the literature that suggests the existence of a critical nucleus (of 12 dimers) that forms via multi-step process before the nucleation begins. In principle, one could write a simpler model with a single step nucleation reaction only, if the assumption of a critical nucleus of size 12 dimers is relaxed. For example, if we assume critical nucleus requires only 3 ($p=3$) dimers, then the second equation describing nucleus formation can be discarded (then S_b will be N_{nuc}). However, we decided to keep the critical nucleus size to 12 to be in agreement with the existing models and, hence, use a two-step process for nucleation.

This choice also makes sense biologically because it is reasonable to expect that the size of the nucleus should be similar to a full ring at the base of the microtubule. The supplementary text on mathematical modelling describes these assumptions and the rationale behind different terms in the model.

Minor issue

- Although unexplained, it is known that all cells are not equally permeant to SiR-tubulin. I am wondering whether the non-tensed shape of the mutants could prevent the entry of SiR-tubulin at specific stages of sporozoites formation.

A: This is indeed a possibility. However, the observation that we can correlate the number of microtubules as found by EM with the staining intensity obtained by SiR tubulin would argue that permeability to SiR tubulin is mostly unaffected.

- could microtubules formed with alpha2-tubulin be more dynamic (and thus compatible with mitosis) than those formed with alpha1-tubulin?

A: This is a possibility indeed. In addition to the parasite lines described in this paper, we have also made one where alpha2-tubulin is replaced by alpha1-tubulin. Like the reverse line, this has had no impact on the blood stages, suggesting that at least for this stage there is no difference in their capacity to undergo mitosis. This line, together with others will be described in a follow-up paper.

Referee #2:

This is an elegant study from the Frischknecht lab that offers fundamental new insight into the role of microtubules in malaria sporozoite formation and infectivity, with implications for cell biology more generally. The study is elaborate and technically challenging, but well conducted and written. The results support the conclusions and are convincing. I have no hesitation in recommending that the paper be accepted for publication.

A: Many thanks for the enthusiastic support.

The authors have not mentioned an important structure of the pellicle, the subpellicular network, that links the subpellicular microtubules and inner membrane complex. Alveolin and other protein components of the SPN contribute to sporozoite shape, size, tensile strength, motility and infectivity, and can affect the distribution of the microtubules. Changing the size and number of microtubules might impact on SPN formation or function, and thus I feel it would be appropriate that a section is devoted to this in the Discussion.

A: Thanks for pointing out this oversight. Indeed the involvement of the SPN in sporozoite formation is one topic we would like to address experimentally in the future. We added a paragraph in the discussion (lines 507-516) including some new references. The text reads as follows:

Another possibility contributing to decreased infectivity of the parasites showing fewer microtubules could be a change in the subpellicular network, a cytoskeletal system underlying the inner membrane complex and being linked to microtubules (Gould *et al.*, 2008; Kudryashev *et al.*, 2012; Harding & Meissner 2014, Frischknecht & Matuschewski 2017). It is not clear how this network is established but perturbation by genetic deletion of proteins shows that a destabilization of the network influences sporozoite shape and tensile strength (Khater *et al.*, 2004). It is possible that the subpellicular network interconnects with the microtubules during parasite formation and is affected by a perturbation of microtubules. Indeed the opposite has been shown recently through deletion of the subpellicular network protein G2 in *P. berghei* sporozoites (Trempe *et al.*, 2013) and GAPM1a in *T. gondii* (Harding *et al.*, 2019).

Referee #3:

Plasmodium species encode two α -tubulins (-1 and 2). PlasmoDB data indicate that at least in *P. falciparum*, they are both expressed at all stages of development, with particularly high expression in oocysts. This manuscript by Spreng *et al.* describes an analysis of the consequences of deletion of the α -tubulin genes. The authors find that α -tubulin-2 is essential, as reported previously (Kooij *et al.* 2005). The authors are successful in their attempt to delete α -tubulin-1. Previous attempts to knock out the gene for α -tubulin-1 in *P. berghei* have been unsuccessful (Kooij *et al.* 2005). The authors should comment on why they have had success where previous attempts have failed.

A: It is correct that both isoforms are expressed at all stages. Expression of proteins across the life cycle is not completely conserved between species. A similar situation has been reported by us and our collaborators before for the actin binding protein coronin, which is expressed in blood stages in *P. falciparum* but not in *P. berghei* (Olshina *et al.*, Malaria J 2015 and Bane *et al.* PLoS Pathogens 2016). In a sense we were lucky to study *P. berghei* to address the role of microtubules in sporogony. Our success is most likely being rooted in improved transfection methods and we now state this in line 124 - 125.

The authors found that the α -tubulin-1(-) parasites showed no defect in blood stage development nor in the initial stages of mosquito infection. They report that in the α -tubulin-1(-) parasites, microtubules are not present in sporozoites developing within the oocysts and that sporozoites do not develop correctly.

It is not clear what is happening with α -tubulin-2 in these parasites. Previous work shows that alpha-tubulin-2 is transcribed in the asexual blood stages, gametocytes, ookinetes and oocysts (Kooij et al 2005). The authors undertook qRT-PCR analysis of α -tubulin-2 in oocysts and found a low level of expression, which was unchanged in the α -tubulin-1(-) parasites.

Which α -tubulin isoforms are expressed in asexual blood stages and in gametocytes?

A: Both α -tubulin isoforms are expressed in asexual blood stages, although from data by Otto et al., 2014 alpha2-tubulin is much stronger expressed. This is now stated in lines 118-120.

Is α -tubulin-2 expressed? Does it perform the tubulin functions in these stages?

A: Yes, the the absense of α -tubulin-1 fully functioning gametes and ookinetes are formed

Why is it not observed by SiR-tubulin in early stage the α -tubulin-1(-) oocysts where it is expressed?

A: We do report a SiR-tubulin staining in early oocysts, see Figure 3B. We have now also included EM images to show that these SiR tubulin foci likely correspond to spindles and hemispindels in early oocysts (see new EV Figure 4).

Can it be confirmed that alpha-tubulin-2 binds SiR-tubulin (cf yeast tubulin which does not).

A: Yes, if we force a2 expression in sporozoites we readily detect MTs by SiR tubulin, see figure 5 and supplementary figure 6.

Based on their studies with the early stages of nuclear division in oocysts the authors conclude that the chromosome segregation needed for nuclear division occurs even in the absence of microtubules. What is driving chromosomal segregation during asexual division?

A: As discussed in line 424-447 we think that it might be possible, like in fission yeast, that actin plays a role. Naturally this needs much more work to be understood.

The authors need to present Western and qRT-PCR analysis of alpha and beta tubulins in early in the early oocyst stage as well as SiR-tubulin labelling and EM data in the asexual stage before it can be concluded that chromosome segregation can occur in the absence of microtubules.

A: As we discussed (lines 314-317) Western analysis is impossible due to tubulins from mosquito tissue. We did perform qRT from as early as day 5 onwards (EV Figure 3). To perform analysis earlier makes less sense as the earliest oocysts are only formed around day 3. We reformulated that chromosome segregation in oocysts can occur without MTs, such that it becomes clearer that during early oocyst formation they appear to play a role (as suggested by Figure 3B and the new EV Figure 4) but not in later divisions. To this end we added statement (marked yellow) in lines 316, 424, 443 and 450-451. We also included images of SiR tubulin staining as well as electron micrographs showing both subpellicular and nuclear microtubules of blood stages in EV Figure 4A. These show similar staining (SiR tubulin) and microtubules (EM) in WT and alpha1 knock-out parasites suggesting that alpha2 tubulin is sufficient for proper segregation in blood stages. We also expanded the materials to accommodate this new work (lines 700-706 and 717).

It appears more likely that α -tubulin-2 performs the various functions of microtubules at all stages of development until the later stages of nuclear division in oocysts, where the levels become insufficient. A careful analysis of α -tubulin-1 and -2 at the protein level would be needed to determine whether α -tubulin-2 is contributing to the division process.

A: We agree but as discussed (lines 314-317) this is technically not possible. We attempted both Western blotting and proteomic analysis. We added “too much contamination from mosquito midgut tubulin” to the discussion. One way to do this would be to engineer a mutant that expresses after a skip peptide mCherry after a2 tub and GFP after a1 tub. We might include this for our follow up study on the role of a2.

The authors examined the effect of replacing wildtype α -tubulin-1 with a codon-modified α -tubulin-1 that is expressed at a lower level. Interestingly, they found that the lower level of expression was

associated with the formation of fewer microtubules that exhibit a similar length to the MTs in wildtype parasites. Similarly, other modifications to α -tubulin-1, such as adding the C-terminal residues from α -tubulin-2 resulting in lower levels of expression and fewer (and in some cases shorter) microtubules. The defects in MT production led to defects in ability of transfer to the salivary glands and to transmit to recipient mice. It is perhaps not surprising that alterations in the levels of expression of tubulins leads to aberrant biology.

A: Indeed it is not, but what amazed us is that we could titrate the numbers of MTs by the levels of expression, a first not just in Plasmodium but in any system. Hence, this shows the value of protozoans to study fundamental questions in cell biology.

The authors developed a mathematical model that supports the conclusion that the concentration of α -tubulin protomers severely limits the rate of nucleation leading to fewer MTs that have a similar length.

In summary, this is a very interesting study but additional work is needed to support the authors conclusions.

A: We thank the reviewer for his constructive critique and hope that our answers and the newly included EV Figure 4 suffice to proceed with publication of our manuscript.

2nd Editorial Decision

28th Mar 2019

Thank you for submitting a revised version of your manuscript. It has now been seen by all of the original referees whose comments are shown below.

As you will see, the referees find that all criticisms have been sufficiently addressed and recommend the manuscript for publication. However, before I can send the official acceptance letter, there are a few editorial issues concerning text and figures that I need you to address.

REFeree REPORTS:

Referee #1:

the authors have properly respond to my question and made the appropriated changes to their manuscript which I look forward seeing published in EMBO Journal.

Referee #2:

The authors have adequately addressed my comments from the first round of revision and I am happy for this manuscript to be accepted for publication.

Referee #3:

The authors have addressed my queries.

2nd Revision - authors' response

12th Apr 2019

The authors performed the requested editorial changes.

Thank you for submitting your revised manuscript. I have now looked at everything and all looks fine. Therefore I am very pleased to accept your manuscript for publication in The EMBO Journal.

Congratulations on the very nice work!

Corresponding Author Name: Friedrich Frischknecht

Manuscript Number: 100984